

# Hydrodynamics with triangular point group

Aaron J. Friedman[1⋆○], Caleb Q. Cook[2†○] and Andrew Lucas[1‡]

**1** Department of Physics and Center for Theory of Quantum Matter,
University of Colorado, Boulder CO 80309, USA
**2** Department of Physics, Stanford University, Stanford CA 94305, USA

⋆ aaron.friedman@colorado.edu , † calebqcook@gmail.com ,
‡ andrew.j.lucas@colorado.edu

## Abstract

When continuous rotational invariance of a two-dimensional fluid is broken to the discrete, dihedral subgroup $D_6$ – the point group of an equilateral triangle – the resulting anisotropic hydrodynamics breaks both spatial-inversion and time-reversal symmetries, while preserving their combination. In this work, we present the hydrodynamics of such $D_6$-symmetric fluids, identifying new symmetry-allowed dissipative terms in the hydrodynamic equations of motion. We propose two experiments – both involving high-purity solid-state materials with $D_6$-invariant Fermi surfaces – that are sensitive to these new coefficients in a $D_6$-invariant electron fluid. In particular, we propose a local current imaging experiment (which is present-day realizable with nitrogen vacancy center magnetometry) in a hexagonal device, whose $D_6$-exploiting boundary conditions enable the unambiguous detection of these novel transport coefficients.



## Contents

○ These authors contributed equally to the development of this work.

# 1 Introduction

Recent years have seen numerous experimental realizations of viscous electron flow on micron lengthscales in high-quality solid-state devices. While the most compelling evidence arises in materials with small Fermi surfaces, such as graphene [1–9] or GaAs [10, 11], anisotropic materials such as $PdCoO_2$ [12], $PtSn_4$ [13] or $WTe_2$ [14] may also display hydrodynamic behavior (see [15] for a review). Independent of whether these particular materials realize hydrodynamic electron flows, the question of what types of hydrodynamic signatures may realize in anisotropic materials with discrete point groups holds substantial interest for experimental and theoretical physics.

Such anisotropic hydrodynamics—which manifestly break Galilean invariance—-are most naturally realized in electron fluids. In general, such anisotropy results from the particular point group of the underlying lattice, which constrains the resulting Fermi surface. Because electron hydrodynamics is predominantly controlled by scattering near the Fermi level, the equations of motion are sensitive to the symmetry group realized by the Fermi surface. With only a handful of exceptions [16–21], most of the theoretical literature on electron hydrodynamics restricts to *isotropic* liquids [22–30], which have continuous rotational invariance. However, fluids with discrete (finite) rotational point groups may realize hydrodynamic phenomena not possible in isotropic fluids.

In this work, we consider fluids with discrete rotational symmetry *and* broken spatial inversion symmetry,[1] focusing on two spatial dimensions. However, we note that the hydrodynamic behavior realized in any material with an inversion-breaking point group in general must be analyzed on a case-by-case basis. In this work, we consider the hydrodynamics of fluids with threefold rotational symmetry and reflection symmetry—e.g., electron fluids in materials whose Fermi surfaces have the symmetries of an equilateral triangle (see, e.g., Fig. 2). The corresponding point group is known as "32" in crystallographic notation, or $D_6$ (the dihedral group of order six) in mathematical notation. We use the latter notation herein (see Tab. 1). Our primary motivation for examining $D_6$-invariant hydrodynamics is that $D_6$ is the simplest (i.e., smallest) point group that (*i*) is naturally realized in two-dimensional crystals and (*ii*) breaks spatial inversion symmetry. In this sense, $D_6$ is a natural choice of point group to investigate the effect of inversion breaking on hydrodynamics in the presence of discrete rotation symmetry.

The families of microscopic Hamiltonians compatible with such triangular point group are detailed in Sec. 2.2, where we also discuss the roles of inversion and time-reversal symmetries. Our goal is to understand what hydrodynamic consequences follow from the explicit breaking of inversion symmetry upon reducing from continuous to discrete rotational symmetry. For example, do new terms appear in the constitutive relations for the current and stress tensor? Can new hydrodynamic phenomena arise that would be forbidden in isotropic fluids?

Answering these questions affirmatively, we show that a Fermi liquid with $D_6$ point group exhibits one new hydrodynamic coefficient that—to the best of our knowledge—has not yet been identified in the literature, outside of a technical companion piece to this paper [34]. One simple consequence of this new hydrodynamic coefficient is that placing the electron fluid in a background electric field $E$ produces shear stresses $\tau_{xy} = \tau_{yx}$ or $\tau_{xx} = -\tau_{yy}$ proportional to $E$. This is reminiscent of the classical theory of piezoelectricity; in the context of the mechanics of solids with $D_6$ point group, such a term relating stress to electric field indeed appears in the constitutive relations. However, in the setting of electron fluids, the shear stress need not be associated with a mechanical deformation of the ions, but rather with the (approximately) conserved momentum of the electrons moving inside a fixed crystalline lattice. Put more formally, in a piezoelectric solid, rotational symmetry is spontaneously broken; in the electron liquid, it is explicitly broken. An important consequence of this subtle difference is that the electron fluid's "piezo" effect is *entropy producing* (i.e. dissipative), in contrast to other recently developed theories of anisotropic systems [35, 36].

In Sec. 4 we present two experiments capable of identifying the hydrodynamic signatures of the point group $D_6$. The current-imagine experiment that we propose in Sec. 4.1 is uniquely capable of detecting this new dissipative coefficient in electron liquids using nitrogen-vacancy center magnetometry in a highly symmetric, hexagonal device. Due to the $D_6$-exploiting geometry of the proposed device's boundary conditions (corresponding to the particular arrange-

---

[1]Refs. [31–33] explore systems in which inversion symmetry is broken due to Berry curvature, while preserving isotropy. Our results on the hydrodynamic coefficients allowed by breaking inversion symmetry will therefore differ qualitatively from [31–33].

Table 1: *Left*: Naming conventions for the point group of an equilateral triangle. We refer to this group as $D_6$ throughout. *Right*: Naming conventions for the three irreducible representations of $D_6$ (see Sec. 2.1.2).

| Point group of equilateral triangle | |
|---|---|
| Our notation | $D_6$ |
| Schoenflies | $D_3$ |
| Hermann–Mauguin | 32 |
| Coxeter | $[2,3]^+$ |

| Irreducible representations | |
|---|---|
| Our notation | Crystallographic |
| $U_0^+$ | $A_1$ |
| $U_0^-$ | $A_2$ |
| $R_1$ | $E$ |

ment of current-carrying leads affixed to the sample), a current signal appears at the device center *only* if the hydrodynamic theory is invariant under $D_6$, but no signal appears if the theory has *any* larger point group (such as that of a hexagon, $D_{12}$). In contrast, in Sec. 4.2 we show that inversion-breaking effects are not as effectively probed by simpler, more conventional Hall-effect experiments in narrow channels [37] .

Finally, in Sec. 5 we confirm using kinetic theory that in microscopically plausible models of anisotropic Fermi surfaces, the new dissipative coefficients that we predict are indeed present, and that their magnitudes are not unexpectedly small. We also state the temperature dependence of these coefficients and give predictions for their value in the familiar example of (half of) ABA trilayer graphene. As a result of these predictions, we expect that the $D_6$-invariant hydrodynamics we derive herein may be discernible in near-term experiments.

## 2 Symmetries and microscopic models

The hydrodynamic equations governing the dynamics of a fluid are constrained by the irreducible representations (irreps) of its symmetry group. In this section, we summarize a few preliminary facts that will play an important role in developing the symmetry-constrained hydrodynamics of $D_6$-invariant fluids in subsequent sections. In general, we consider the hydrodynamics of electron fluids, where the point group inherits from the underlying lattice structure and is reflected by the dispersion relation $\varepsilon(\boldsymbol{p})$ (2.14).

We first consider the representation of the continuous symmetry O(2) (corresponding to isotropic fluids) before discussing the particulars of the subgroup $D_6 \subset O(2)$. In particular, we emphasize the importance of the fact that this dispersion breaks both spatial-inversion and time-reversal symmetries while preserving their combination.

### 2.1 Point group symmetry

In this work, we investigate two-dimensional fluids whose point group is generated by (*i*) a discrete, threefold rotation $\rho$ and (*ii*) a reflection $r$ about a fixed line passing through the rotation center, subject to the relations $r\rho r\rho = \rho^3 = r^2 = e$, where $e$ is the identity element. This is the group of planar symmetries of an equilateral triangle (three orientation-preserving rotations $\rho^k$ and three orientation-reversing rotations $r\rho^k$, where $k \in \{0,1,2\}$), which we denote by $D_6$ (the dihedral group of order six). We denote this point group by $D_6$ to emphasize the number of symmetry elements it contains. Elsewhere in the literature, this point group is sometimes denoted as "$D_3$", which instead emphasizes the number of sides of the symmetry polygon (i.e., a triangle). The point group of an equilateral triangle is also known as "32" in Hermann–Mauguin (or "international") notation [38]. The various nomenclature conventions are summarized in Tab. 1; in the remainder, we refer to this point group exclusively as "$D_6$."

Note that $D_6$ is a subgroup of the two-dimensional orthogonal group O(2)—the group of planar symmetries of a circle. As a result, O(2) contains not only spatial reflections, but also *continuous* rotations by arbitrary angles. Isotropic fluids (such as the electron fluid in graphene near charge neutrality [15]) have point group O(2).

Importantly, we show that there exists a rank-three tensor $\lambda_{ijk}$ that—despite transforming nontrivially under the full isotropic point group O(2)—is *invariant* under the subgroup $D_6$. This implies that breaking rotational invariance from the continuous group O(2) to the discrete subgroup $D_6$ generates terms in the hydrodynamic expansion proportional to $\lambda_{ijk}$ that are disallowed under O(2). Elucidating the hydrodynamic implications of the new $D_6$-invariant tensor $\lambda_{ijk}$ is one of the primary goals of this work.

### 2.1.1 O(2) representation theory

The isotropic group O(2) has two (real) one-dimensional irreps $\mathcal{U}_0^{\pm}$ and infinitely many (real) two-dimensional irreps $\mathcal{R}_k$ (for $k \in \mathbb{N}$) [16]. The one-dimensional irreps $\mathcal{U}_0^{\pm}$ correspond to mathematical objects that are trivial under O(2) rotations and are even ($\mathcal{U}_0^+$) or odd ($\mathcal{U}_0^-$) under spatial reflection; we refer to these as the "scalar" and "pseudoscalar" irreps, respectively. The two-dimensional representations $\mathcal{R}_k$ correspond to mathematical objects that rotate by angle $k\theta$ under O(2) rotations by angle $\theta$; we refer to $\mathcal{R}_k$ as the "spin-$k$" irrep, and $\mathcal{R}_1$ in particular as the "vector" (or "spin-one") irrep. The hydrodynamic equations of motion are constrained by consideration of these irreps.

Since we develop the hydrodynamic expansion in tensorial form, we now consider how these representations are realized in vector spaces of tensors. The group O(2) has a natural action on two-dimensional, rank-$n$ tensors of the form $T_{i_1\cdots i_n}$ with $i_\alpha \in \{1, 2\}$; this action is found by independently transforming each index $i_\alpha$ according to the irrep $\mathcal{R}_1$, which may be thought of as "rotating each index as a vector" [16]. In particular, this means that rank-$n$ tensors transform under O(2) according to an $n$-fold tensor product of $\mathcal{R}_1$, i.e.,

$$T_{i_1\cdots i_n} \in \mathcal{R}_1^{\otimes n}. \tag{2.1}$$

Importantly, the product representation $\mathcal{R}_1^{\otimes n}$ is generally *reducible*—i.e., it can be rewritten as a direct sum of O(2) irreps. For example, a generic rank-two tensor $T_{ij}$ transforms under O(2) as

$$T_{ij} \in \mathcal{R}_1 \otimes \mathcal{R}_1 = \mathcal{U}_0^+ \oplus \mathcal{U}_0^- \oplus \mathcal{R}_2, \tag{2.2}$$

where the final decomposition can be seen explicitly using the tensor identity

$$T_{ij} = \underbrace{\left(\frac{\delta_{kl}}{\sqrt{2}} T_{kl}\right)\frac{\delta_{ij}}{\sqrt{2}}}_{\in \mathcal{U}_0^+} + \underbrace{\left(\frac{\epsilon_{kl}}{\sqrt{2}} T_{kl}\right)\frac{\epsilon_{ij}}{\sqrt{2}}}_{\in \mathcal{U}_0^-} + \underbrace{\left(\frac{\sigma_{kl}^x}{\sqrt{2}} T_{kl}\right)\frac{\sigma_{ij}^x}{\sqrt{2}} + \left(\frac{\sigma_{kl}^z}{\sqrt{2}} T_{kl}\right)\frac{\sigma_{ij}^z}{\sqrt{2}}}_{\in \mathcal{R}_2}, \tag{2.3}$$

which is simply the familiar statement that, under the action of O(2), rank-two tensors decompose into a scalar trace component $\in \mathcal{U}_0^+$, a pseudoscalar antisymmetric component $\in \mathcal{U}_0^-$, and a spin-two traceless symmetric component $\in \mathcal{R}_2$, corresponding to projection onto the irreducible subspaces. For general tensor products of O(2) spin-$k$ irreps $\mathcal{R}_k$, the general decomposition is given by [16],

$$\mathcal{R}_k \otimes \mathcal{R}_l = \mathcal{R}_{|k-l|} \oplus \mathcal{R}_{k+l}, \tag{2.4}$$

where we have defined the representation $\mathcal{R}_0 \equiv \mathcal{U}_0^+ \oplus \mathcal{U}_0^-$, which essentially follows from the trigonometric identity

$$(\cos k\theta) \cdot (\cos l\theta) = \frac{1}{2}\cos[(k-l)\theta] + \frac{1}{2}\cos[(k+l)\theta], \tag{2.5}$$

$$v^3 = \left(-\frac{\sqrt{3}}{2}s, \frac{1}{2}s\right) \qquad v^2 = \left(+\frac{\sqrt{3}}{2}s, \frac{1}{2}s\right)$$

$$v^1 = (0, -s)$$

Figure 1: An equilateral triangle is shown, with vectors $v^{1,2,3}$ connecting the triangle center to each side midpoint. The $D_6$-invariant, rank-three tensor $\lambda_{ijk}$ can be defined by symmetrizing the threefold tensor product $v_i^1 v_j^2 v_k^3$ over $D_6$; see (2.10) for the corresponding formula. For numerical convenience, we take the overall scale $s$ of the triangle (which does not affect the invariance of $\lambda_{ijk}$) to be $s = 4^{1/3}$.

### 2.1.2 $D_6$ representation theory

The group $D_6$ has two one-dimensional irreps $U_0^\pm$ and one two-dimensional irrep $R_1$ [16]. The one-dimensional irreps $U_0^\pm$ correspond to mathematical objects that are trivial under $D_6$ rotations and "parity even" ($U_0^+$) or "parity odd" ($U_0^-$) under the $D_6$ reflection. As the $D_6$ irreps $U_0^\pm$ are trivial under rotations, they may be identified with the $O(2)$ irreps $\mathcal{U}_0^\pm$ when the latter are *restricted* to $D_6$ via

$$\mathcal{U}_0^\pm\big|_{D_6} = U_0^\pm, \tag{2.6}$$

where $U$ corresponds to $D_6$ and $\mathcal{U}$ to $O(2)$. We then identify the two-dimensional (or "vector") irrep of $D_6$ as $R_1$, corresponding to mathematical objects that transform as vectors under discrete rotations by integer multiples of $(2\pi/3)$. As with the one-dimensional irreps $U_0^\pm$, $R_1$ recovers upon restricting the vector irrep $\mathcal{R}_1$ of $O(2)$ to $D_6$ via

$$\mathcal{R}_1\big|_{D_6} = R_1. \tag{2.7}$$

The fate of higher, spin-$k$ $O(2)$ irreps $\mathcal{R}_k$ upon restriction to the subgroup $D_6$ can be derived using representation theory [16]. Instead, however, we motivate the result by imagining spin-$k$ irreps $\mathcal{R}_k$ acting on an equilateral triangle. For example, it is easy to see that $\mathcal{R}_2\big|_{D_6} = R_1$, since a clockwise (counterclockwise) rotation of the triangle by $2 \times (2\pi/3)$ is equivalent to a counterclockwise (clockwise) rotation of the triangle by $(2\pi/3)$. Similarly, we can see that $\mathcal{R}_3\big|_{D_6} = \mathcal{R}_0\big|_{D_6} = U_0^+ \oplus U_0^-$ since rotations of $3 \times (2\pi/3) = 2\pi$ leave the triangle invariant. A slight generalization of these observations to generic spin-$k$ leads to the conclusion

$$\mathcal{R}_k\big|_{D_6} = \begin{cases} U_0^+ \oplus U_0^-, & k \bmod 3 = 0, \\ R_1, & \text{otherwise}, \end{cases} \tag{2.8}$$

and we can now derive the action of $D_6$ on rank-three tensors; using (2.1), (2.4), and (2.8), we find

$$T_{ijk}\big|_{D_6} \in \mathcal{R}_1 \otimes \mathcal{R}_1 \otimes \mathcal{R}_1\big|_{D_6} = 3\mathcal{R}_1 \oplus \mathcal{R}_3\big|_{D_6} = U_0^+ \oplus U_0^- \oplus 3R_1. \tag{2.9}$$

Crucially, this decomposition contains a $U_0^+$ subspace, which implies that rank-three tensors—which posses no scalar component under the isotropic group $O(2)$—do, in fact, have a scalar component in $D_6$. In other words, there exists a rank-three tensor $\lambda_{ijk}$ that is invariant under $D_6$ but *not* $O(2)$.

We now explicitly derive the $D_6$-invariant tensor $\lambda_{ijk}$ in an intuitive way. Consider the equilateral triangle depicted in Fig. 1, with vectors $v^{1,2,3}$ connecting the triangle's center to

the midpoints of its sides. If we fully symmetrize the threefold tensor product $v_i^1 v_j^2 v_k^3$ over the group $D_6$, it is clear that the resulting rank-three tensor must be invariant under the symmetries of the triangle, i.e. lie in the $U_0^+$ subspace of $\otimes^3 \mathcal{R}_1$. Carrying out this procedure yields

$$\lambda_{ijk} = \frac{1}{3!} \sum_{g \in D_6} v_i^{g \cdot 1} v_j^{g \cdot 2} v_k^{g \cdot 3} = \delta_{k1} \sigma_{ij}^x + \delta_{k2} \sigma_{ij}^z, \tag{2.10}$$

where triangle side $i$ is sent to triangle side $g \cdot i$ under the $D_6$ group operation $g$ (see Fig. 1), and $\sigma^a$ are Pauli matrices; note that the final equality above can be verified by direct computation. Even more explicitly, the tensor $\lambda_{ijk}$ satisfies

$$\lambda_{112} = \lambda_{121} = \lambda_{211} = -\lambda_{222} = 1, \tag{2.11}$$

with all other components zero. Note that $\lambda_{ijk}$ is fully symmetric (i.e., invariant under exchange of any two indices).

## 2.2 Time reversal and inversion symmetries

Another key observation is that an electron fluid with a triangular Fermi surface breaks time-reversal symmetry [39]. A simple way to see this is as follows: Consider a microscopic Hamiltonian describing $N$ interacting electrons with canonical positions and momenta $\boldsymbol{x}_i$ and $\boldsymbol{p}_i$, respectively. Under time reversal (denoted by $\Theta$) one has

$$\Theta \cdot (\boldsymbol{x}_i, \boldsymbol{p}_i) = (\boldsymbol{x}_i, -\boldsymbol{p}_i). \tag{2.12}$$

The effective Hamiltonians describing systems with "triangular" dispersion relations are of the (first-quantized) form

$$H = \sum_{i=1}^{N} \varepsilon(\boldsymbol{p}_i) + \sum_{i<j} V_{ij}(\boldsymbol{x}_i - \boldsymbol{x}_j), \tag{2.13}$$

where $V_{ij}$ is a two-body interaction and the dispersion $\varepsilon(\boldsymbol{p})$ is some function of $p_x$ and $p_y$ with $D_6$ invariance. One such family of $D_6$-invariant dispersion relations $\varepsilon$ is given by

$$\varepsilon(\boldsymbol{p}) = a\left(p_x^2 + p_y^2\right) + b\left(3p_x^2 - p_y^2\right)p_y + c\left(p_x^2 + p_y^2\right)^2, \tag{2.14}$$

and contour plots (i.e. Fermi surfaces) corresponding to this dispersion relation are plotted for various values of $a$, $b$, and $c$ in Fig. 2. In general, the precise dispersion relation (2.14) may contain other $D_6$-symmetric corrections, and inherits from the underlying lattice structure. The manner in which this effects the hydrodynamics is captured at the microscopic level in our consideration of kinetic theory in Sec. 5.

Clearly, and independently of the microscopic model being considered, $\varepsilon(-\boldsymbol{p}) \neq \varepsilon(\boldsymbol{p})$, manifestly breaking time-reversal symmetry. As a result, the Hamiltonian itself breaks time-reversal symmetry:

$$H(\Theta \cdot (\boldsymbol{x}, \boldsymbol{p})) \neq H(\boldsymbol{x}, \boldsymbol{p}). \tag{2.15}$$

However, there *is* a sense in which time-reversal symmetry is restored in such systems. Consider the spatial inversion operation (denoted by $\mathcal{I}$):

$$\mathcal{I} \cdot (\boldsymbol{x}_i, \boldsymbol{p}_i) = (-\boldsymbol{x}_i, -\boldsymbol{p}_i), \tag{2.16}$$

where we emphasize that $\mathcal{I}$ is *not* an element of $D_6$. As with time reversal, we find for inversion that

$$H(\mathcal{I} \cdot (\boldsymbol{x}, \boldsymbol{p})) \neq H(\boldsymbol{x}, \boldsymbol{p}), \tag{2.17}$$

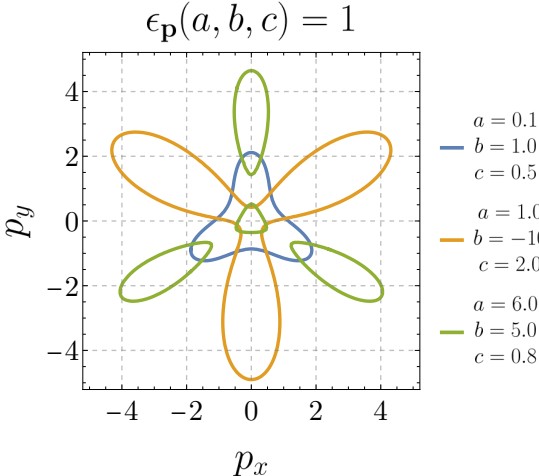

Figure 2: Contour plots of the $D_6$-invariant dispersion relation $\varepsilon$ (2.14). Fermi surfaces of this type have the same point symmetry as an equilateral triangle—i.e., discrete threefold rotational invariance and an in-plane axis of reflection symmetry. Note in particular that these Fermi surfaces lack $\boldsymbol{p} \to -\boldsymbol{p}$ invariance and hence independently break both spatial-inversion and time-reversal symmetries, while nevertheless retaining their combination as a symmetry.

due to the dispersion relation. However, assuming that the interaction between particles is inversion symmetric (i.e., $V(-\boldsymbol{x}) = V(\boldsymbol{x})$), we have that

$$H(\mathcal{I}\Theta \cdot (\boldsymbol{x}, \boldsymbol{p})) = H(-\boldsymbol{x}, \boldsymbol{p}) = H(\boldsymbol{x}, \boldsymbol{p}), \tag{2.18}$$

and we conclude that, in addition to the $D_6$ point group symmetry, the electron liquid also has an "IT symmetry," corresponding to the combination of spatial inversion and time reversal. This IT (or $\mathcal{I}\Theta$) symmetry turns out to play an important role both in developing the hydrodynamic expansion and in constraining what types of experiments can detect phenomena unique to liquids with $D_6$ point group.

## 3 Hydrodynamics

We now describe the hydrodynamics of a two-dimensional fluid with triangular point group. For simplicity, we assume that the conserved quantities are particle number (or charge or mass, with density $\rho$) and momentum (with density $\pi_j$). This choice is sufficient to elucidate all of the interesting structure that can arise without carrying around additional coefficients; the mathematical structure of our theory would hold just as well if $\rho$ corresponded to energy density, as may be appropriate for a phonon fluid [40, 41]. We also note that in an electronic Fermi liquid, under most circumstances, it is a good approximation to neglect energy conservation since each quasiparticle carries the same (Fermi) energy with it, on average [15].

### 3.1 Continuity equations and constitutive relations

Hydrodynamics is an effective theory describing the long-wavelength and late-time physics of interacting systems. Most observables in many-body systems are "fast" degrees of freedom, in that they quickly decay to zero, leaving behind only the "slow" modes, which dominate at late times and over long distances. In generic chaotic systems, the latter always correspond to conserved quantities; their slow dynamics corresponds to the slow transport of conserved

quantities over long lengthscales. Hydrodynamics integrates out the "fast" physics to provide a useful, "effective" description of the relaxation of these conserved quantities from local to global equilibrium (see Sec. 5).

For the systems relevant to this work, the relevant slow modes (i.e., conserved quantities) correspond to charge (or particle) density $\rho$ and momentum density $\pi_j$.[2] It is conventional to replace some of these densities with their thermodynamic conjugate variables: For $\rho$, this is chemical potential $\mu$, while for momentum density $\pi_j$, the conjugate is the velocity $v_j$. By convention, we take the hydrodynamic modes to be $\rho$ and $v_j$.

The continuity equations require that any change in the local density of some conserved charge (i.e., $\partial_t \rho$) within an infinitesimal fluid volume be offset by a flux of the current corresponding to that charge out of the volume, i.e.,

$$\partial_t \rho + \partial_i j_i = 0, \tag{3.19a}$$

$$\partial_t \pi_i + \partial_i \tau_{ij} = 0. \tag{3.19b}$$

The current corresponding to a conserved density is written in terms of the conserved charges (the slow modes described by hydrodynamics) using constitutive relations. Essentially, one writes down all symmetry-allowed combinations of the hydrodynamic fields (density and velocity, in this case) and their derivatives to leading order.

We also work in the linear-response regime, where we take $\rho(x, y, t) \rightarrow \rho_0 + \delta\rho(x, y, y)$ and $v_j(x, y, t) \rightarrow \delta v_j(x, y, t)$, and only keep terms to linear order in $\delta$. The quantity $\rho_0 = (\partial p/\partial \mu)_{\text{eq}}$ (where $p$ is the pressure and $\mu$ the chemical potential, and "eq" denotes that the righthand side is evaluated with $\delta\rho = 0$) is the "background" charge density, and is fixed by thermodynamic relations; the "background" velocity is taken to be zero.

We now construct the current $j_i$ and stress tensor $\tau_{ij}$ via derivative expansion. The derivatives must act on the hydrodynamic variables $\rho$ and $v_i$ (the charge density and velocity). We note that a frame can always be chosen such that no time derivatives appear in the expansion, leaving only powers of $\partial_i$. Additionally, the only tensor objects that can appear in the expansion are those allowed by $D_6$ representation theory, as discussed in Sec. 2.1.2. Those $D_6$-invariant tensors are

$$\delta_{ij}, \quad \epsilon_{ij}\epsilon_{kl} = \delta_{ik}\delta_{jl} - \delta_{il}\delta_{jk}, \quad \lambda_{ijk}. \tag{3.20}$$

The hydrodynamic equations of motion correspond to the conservation laws (3.19) combined with the constitutive relations, which relate $j_i$ and $\tau_{ij}$ to $\rho$ and $\pi_i$ (or its thermodynamic conjugate, the velocity $v_i$). We first write the most general constitutive relation for the current, allowing all terms that are compatible with $D_6$ symmetry:

$$j_i = \rho_0 v_i + K\lambda_{ijk}v_j v_k - D\partial_i \rho - \alpha\lambda_{ijk}\partial_j v_k + \gamma\lambda_{ijk}\partial_j\partial_k\rho + \theta_1\partial_j\partial_j v_i + \theta_2\partial_i\partial_j v_j + \cdots, \tag{3.21}$$

which we truncate to first order in derivatives. This is sufficient to obtain a linearly stable theory of hydrodynamics in which all finite-wavelength modes decay; hence, we ignore the $\gamma$ and $\theta$ terms above. We also ignore the $K$ term, as it is only nonzero at $O(v^2)$, and therefore vanishes in the linear-response regime of interest. The coefficient $D$ is referred to as the "incoherent diffusion constant," where "incoherent" refers to the fact that $D > 0$ is possible even when momentum is exactly conserved. This effect arises from the fact that, without Galilean invariance, the charge current contains a term that is not proportional to momentum and can relax (see, e.g., [16]).

---

[2]Including energy conservation is straightforward, and is discussed in [34]. For most experimental setups, energy conservation does not qualitatively modify the observable hydrodynamics of an electronic Fermi liquid [15], and so we neglect it.

Indeed, we note that Galilean invariance must be explicitly broken in this system. The Galilean symmetry group's algebra can be understood as follows: The center of mass of the fluid $D_i$, the total momentum $P_i$, and the Hamiltonian (energy) $H$ must satisfy the classical Poisson brackets

$$\{D_i, H\} = m^{-1} P_i \,, \tag{3.22}$$

where $m$ is a microscopic mass scale. In any microscopic model, this fixes the dispersion relation to be

$$\varepsilon(\boldsymbol{p}) = \frac{p_x^2 + p_y^2}{2m} \,, \tag{3.23}$$

which is isotropic and has full $O(2)$ (rather than $D_6$) symmetry. While, in principle, it might be possible to realize electron-electron interactions $V(x, x')$ (2.13) that break $O(2)$ down to $D_6$, in general this would (*i*) require a very different kinetic theory than the standard version we consider in Sec. 5 and (*ii*) be extremely difficult to predict (or engineer) in particular materials. By contrast, it is most natural (and conceptually straightforward) if instead the $D_6$ point group manifests in the dispersion relation $\varepsilon(\boldsymbol{p})$ due to the properties of the underlying lattice. As a direct consequence, Galilean invariance is explicitly broken.

We now consider the momentum density and the constitutive relation for its corresponding current, the (rank-two) stress tensor $\tau_{ij}$. The derivative expansion for $\tau_{ij}$ to first order leads to the constitutive relation

$$\tau_{ij} = \delta_{ij} p + K' \lambda_{ijk} v_k - \eta_{ijkl} \partial_k v_l - \beta \lambda_{ijk} \partial_k \rho + \dots \,, \tag{3.24}$$

where $\lambda$ is the $D_6$ invariant tensor[3] defined in (2.11), the viscosity tensor $\eta_{ijkl}$ is given by

$$\eta_{ijkl} = \eta \left( \delta_{ik}\delta_{jl} + \delta_{il}\delta_{jk} - \delta_{ij}\delta_{kl} \right) + \zeta \delta_{ij}\delta_{kl} + \eta_\circ \epsilon_{ij}\epsilon_{kl} \,, \tag{3.25}$$

and $p$ is the thermodynamic pressure. In the linear response regime,

$$p = c^2 \rho \,, \tag{3.26}$$

where $c$ is the speed of sound, and we neglect the "Coulomb pressure" [15], which only modifies the sound modes via $c \to c(\mathbf{k})$ without affecting the incompressible transverse fluid dynamics of interest herein. In particular, we do not expect any meaningful effect due to the Coulomb pressure on the two experiments we propose in Sec. 4.

Consistency with thermodynamics implies that not all of the terms listed above are independent (e.g., $\rho = \partial p / \partial \mu$), which provides further relations between various quantities. Additionally, it turns out that

$$K' = 0 \,, \tag{3.27}$$

in all known models of kinetic theory (see Sec. 5.2).

Taking $K'$ to vanish is also supported by general arguments that there should be a consistent manner by which to couple the effective field theories of electron fluids to curved space [34]. As an aside, we further note that if $K' \neq 0$, then consistency with the second law of thermodynamics then requires that $K \neq 0$ in (3.21) [34]. More recently, it was shown that the existence of such a $K'$ term will require an exotic kind of chiral anomaly [42], which can be found in actual lattice models (but is unlikely to exist in electron fluids).

---

[3]Also note the useful identity $\lambda_{ijk}\lambda_{imn} = \delta_{jm}\delta_{kn} + \delta_{jn}\delta_{km} - \delta_{jk}\delta_{mn}$. This tensor identity shows that in $D_6$, the traceless symmetric part of a tensor corresponds to the $R_1$ irrep; the projection of a tensor $T_{mn}$ onto $R_1$ is given by $T'_{ij} = \lambda_{ijk}\lambda_{kmn}T_{mn}$.

For fluids with point group $D_6$, hydrodynamics is governed by the following pair of lin-earized continuity equations,

$$\partial_t \rho = -\partial_i j_i = -\rho_0 \nabla \cdot v + D\nabla^2 \rho + 2\alpha\, \partial_x \partial_y\, v_x + \alpha \left(\partial_x^2 - \partial_y^2\right) v_y\,, \tag{3.28a}$$

$$\rho_0 \partial_t v_i = -\partial_j \tau_{ji} = -c^2 \partial_i \rho + \eta_{jikl}\partial_j \partial_k v_l + \beta\lambda_{ijk}\partial_j\partial_k\rho\,, \tag{3.28b}$$

corresponding to charge (3.28a) and momentum density (3.28b). The individual velocity equations are given by

$$\rho_0\,\partial_t\,v_x = -c^2\,\partial_x\,\rho + \eta\,\nabla^2\,v_x + \zeta\,\partial_x\left(\nabla\cdot v\right) - \eta_\circ\,\partial_y\left(\nabla\times v\right) + 2\beta\,\partial_x\partial_y\,\rho\,, \tag{3.29a}$$

$$\rho_0\,\partial_t\,v_y = -c^2\,\partial_y\,\rho + \eta\,\nabla^2\,v_y + \zeta\,\partial_y\left(\nabla\cdot v\right) + \eta_\circ\,\partial_x\left(\nabla\times v\right) + \beta\left(\partial_x^2 - \partial_y^2\right)\rho\,, \tag{3.29b}$$

where $\nabla \times v = \partial_i \epsilon_{ij} v_j = \partial_x v_y - \partial_y v_x$ is the curl in two spatial dimensions.

We note that the viscosity tensor (3.25) has the same form as for fluids with $D_{12}$ point group (i.e., the rotational invariance of a hexagon) [16, 17], and that $\eta_\circ \to 0$ in the case of $O(2)$ rotation symmetry; the new dissipative terms compared to fluids with other point groups involve the $D_6$-invariant tensor $\lambda_{ijk}$ with coefficients $\alpha$ and $\beta$, which represent a sort of "hybrid" between a viscosity and an (incoherent) conductivity.

## 3.2 Onsager relations for the $D_6$ coefficients

The Onsager reciprocal relations for $D_6$ fluids follow straightforwardly from standard argu-ments from statistical mechanics. Abstractly, we define $\mu_a$ as the thermodynamic conjugate to the conserved mode $\rho_a$ (e.g., chemical potential $\mu$ is conjugate to charge density $\rho$). We further suppose that the current $j_a$ associated with the conserved density $\rho_a$ takes the form

$$j_a^i = -\sigma_{ab}^{ij}\nabla_j\mu_b\,, \tag{3.30}$$

with summation over $b$ implied.

Fluids with point group $D_6$ possess neither spatial inversion ($\mathcal{I}$) nor time reversal ($\Theta$) in-dependently; rather, the microscopic system of interest is invariant only under the *combination* $\mathcal{I}\Theta$. By demanding consistency with the fluctuation-dissipation theorem, the matrix $\sigma$ of dissi-pative coefficients (including the new $D_6$ coefficients, viscosity, incoherent conductivity,[4] etc.) must satisfy

$$\sigma_{ab}^{ij} = (\mathcal{I}\Theta)_a (\mathcal{I}\Theta)_b\, \sigma_{ba}^{ji}\,, \tag{3.31}$$

where $(\mathcal{I}\Theta)_a = +1$ when $\mathcal{I}\Theta \cdot \rho_a = \rho_a$, and $(\mathcal{I}\Theta)_a = -1$ when $\mathcal{I}\Theta \cdot \rho_a = -\rho_a$ (i.e., these factors encode the parity under IT of the conserved densities labelled $a$ and $b$).

Note that $\sigma$ is best understood as a matrix by grouping the $ia$ and $jb$ indices; $\sigma$ is then block diagonal, with different blocks corresponding to distinct irreps of the symmetry group (here, $D_6$). In ordinary fluids, when $a = \rho$, $ia$ transforms as a vector, while for $a = \pi_j$, "$ij$" transforms as a rank-two tensor. Since vectors and rank-two tensors do not share irreps, $\sigma_{ab}^{ij}$ must be block diagonal, corresponding to incoherent conductivity when $a = b = \rho$ and viscosity (3.25) when $a = b = \pi$. However, in $D_6$ representations, the rank-two tensor contains an $R_1$ index. As a consequence, we anticipate that $\sigma_{\rho\pi_k}^{ij} \propto \lambda_{ijk}$. This coefficient then corresponds to the $\alpha$ term in (3.28a) and $\beta$ term in (3.28b).

By (3.31), we should expect $\alpha$ and $\beta$ to be related. Charge is even under both time reversal and spatial inversion individually, so that $(\mathcal{I}\Theta)_\rho = 1$, while momentum density is odd under both individually, so that $(\mathcal{I}\Theta)_\pi = 1$. The Onsager relation (3.31) then implies that

$$\sigma_{\rho\,\pi_k}^{ij} = \sigma_{\pi_k\rho}^{ji} \equiv \alpha\,\lambda_{ijk}\,, \tag{3.32}$$

---

[4]Note that the incoherent conductivity is proportional to the coefficient $D$ in (3.21); see (3.36).

and now, using the linear-response relation

$$\chi = \frac{\partial \rho}{\partial \mu} = \frac{\rho_0}{c^2},\tag{3.33}$$

we conclude that

$$\alpha = \beta \chi.\tag{3.34}$$

In some manipulations to follow, it will prove useful to define a single $D_6$ coupling, according to

$$\xi = \frac{\alpha}{\rho_0} = \frac{\beta}{c^2},\tag{3.35}$$

which is a lengthscale that can roughly be interpreted as a scattering length for the momentum-conserving (but inversion-breaking) collisions, as we will later see in Sec. 5 when we consider kinetic theory. Comparing $\xi$ to other lengthscales in the system provides a measure of the extent to which this new hydrodynamic coefficient can be detected in experiments—in other words, the relative significance of realizing point group $D_6$. Dimensionless values of the various hydrodynamic coefficients—including $\alpha \sim \beta$—are provided in Sec. 5.4 for a microscopically inspired model of a Fermi surface.

Finally, we note that the second law of thermodynamics requires that the matrix $\sigma$ of dissipative coefficients be positive semidefinite. This in turn implies (*i*) that the viscosity tensor (3.25) is nonnegative ($\eta \geq 0$), (*ii*) that the incoherent conductivity also be nonnegative ($\sigma_0 \geq 0$), and (*iii*) the following inequality:

$$\sigma_0 \eta \geq \alpha \beta \chi, \quad \text{where} \quad \sigma_0 = D \chi,\tag{3.36}$$

which recovers from (5.105) and can be rewritten as $D \eta \geq \rho_0 c^2 \xi^2$.

### 3.3 Quasinormal modes

From the linearized continuity equations (3.28) we can extract the quasinormal modes of this system. In an ordinary fluid these quasinormal modes would correspond to a sound mode coupling longitudinal momentum ($\pi \parallel k$) and density $\rho$, and a diffusive mode for transverse momentum ($\pi \perp k$). Qualitatively, the same thing happens here. In matrix form, those three equations can be written

$$\partial_t \begin{pmatrix} \rho \\ \rho_0 v_x \\ \rho_0 v_y \end{pmatrix} = \tag{3.37}$$

$$\begin{pmatrix} D\nabla^2 & -\partial_x + 2\frac{\alpha}{\rho_0}\partial_x\partial_y & -\partial_y + \frac{\alpha}{\rho_0}\left(\partial_x^2 - \partial_y^2\right) \\ -c^2\partial_x + 2\beta\partial_x\partial_y & \frac{1}{\rho_0}(\eta+\zeta)\partial_x^2 + \frac{1}{\rho_0}(\eta+\eta_\circ)\partial_y^2 & \frac{1}{\rho_0}(\zeta-\eta_\circ)\partial_x\partial_y \\ -c^2\partial_y + \beta\left(\partial_x^2 - \partial_y^2\right) & \frac{1}{\rho_0}(\zeta-\eta_\circ)\partial_x\partial_y & \frac{1}{\rho_0}(\eta+\zeta)\partial_y^2 + \frac{1}{\rho_0}(\eta+\eta_\circ)\partial_x^2 \end{pmatrix} \begin{pmatrix} \rho \\ \rho_0 v_x \\ \rho_0 v_y \end{pmatrix},$$

and taking the Fourier transform (i.e., $\partial_t \rightarrow -i\omega$ and $\partial_j \rightarrow -ik_j$ with $\rho \rightarrow \widetilde{\rho}$), we have

$$(-i\omega \mathbb{1} + M) \begin{pmatrix} \widetilde{\rho} \\ \rho_0 \widetilde{v}_x \\ \rho_0 \widetilde{v}_y \end{pmatrix} = 0,\tag{3.38}$$

where the matrix $M(\mathbf{k})$ takes the form

$$M\left(k_x, k_y\right) = \tag{3.39}$$

$$\begin{pmatrix} D\,k^2 & -\mathrm{i}\,k_x + 2\frac{\alpha}{\rho_0}k_x k_y & -\mathrm{i}\,k_y + \frac{\alpha}{\rho_0}\left(k_x^2 - k_y^2\right) \\ -\mathrm{i}\,c^2 k_x + 2\beta\,k_x k_y & \frac{1}{\rho_0}(\eta + \zeta)k_x^2 + \frac{1}{\rho_0}(\eta + \eta_\circ)k_y^2 & \frac{1}{\rho_0}(\zeta - \eta_\circ)k_x k_y \\ -\mathrm{i}\,c^2 k_y + \beta\left(k_x^2 - k_y^2\right) & \frac{1}{\rho_0}(\zeta - \eta_\circ)k_x k_y & \frac{1}{\rho_0}(\eta + \zeta)k_y^2 + \frac{1}{\rho_0}(\eta + \eta_\circ)k_x^2 \end{pmatrix},$$

and the normal modes correspond to choices of $\omega(\mathbf{k})$ that satisfy

$$\det\left(-\mathrm{i}\,\omega\,\mathbb{1} + M\right) = 0, \tag{3.40}$$

giving the *quasinormal modes*

$$\omega(\mathbf{k}) = \begin{cases} \pm c\,k - \frac{\mathrm{i}}{2\rho_0}\left(\rho_0 D + \eta + \zeta \mp 2\xi\rho_0 c\,\sin(3\theta)\right)k^2 + \ldots, & \text{``sound mode''}, \\ -\frac{\mathrm{i}}{\rho_0}(\eta + \eta_\circ)k^2 + \ldots, & \text{``shear diffusion mode''}, \end{cases} \tag{3.41}$$

up to $O(k^3)$, where $k_x = k\cos(\theta)$ and $k_y = k\sin(\theta)$, so that

$$\sin(3\theta) = \frac{\left(3k_x^2 - k_y^2\right)k_y}{k^3} = \frac{\lambda_{ijk}k_i k_j k_k}{k^3}, \tag{3.42}$$

and we note that stability of the quasinormal modes (3.41) requires that the imaginary part of $\omega$ be negative (to prevent unphysical exponential growth). This is trivially satisfied for the shear diffusion mode (since $\eta$ and $\eta_\circ$ are positive), and for the sound mode, requires that $\rho_0 D + \eta + \zeta \geq 2\rho_0\xi c\,\sin(3\theta)$. The most "dangerous" case is when the RHS of the foregoing inequality is maximal; this corresponds to, e.g., $\theta \sim \pi/6$ and $\xi^2 = \eta D/\rho_0 c^2$ (3.36). Then the stability condition becomes $\zeta/\rho_0 + \left(\sqrt{D} - \sqrt{\eta/\rho_0}\right)^2 \geq 0$, which is trivially satisfied.

The main new feature in (3.41) is the decay rate $\propto \xi\,\sin(3\theta)k^2$, meaning that sound modes preferentially decay in certain directions, with threefold rotational symmetry. This explicitly demonstrates the $D_6$ symmetry of the theory.

## 3.4 Plasmons

We briefly consider how the inclusion of long-range Coulomb interactions [43, 44] affects the quasinormal modes (3.41) of the $D_6$ fluid.[5] The effects of generic forces can be incorporated into the continuity equations (3.28) via the transformation $\partial_j \mu \to \partial_j \mu - F_j^{\text{ext}}$. Using the fact that $\delta\rho/\delta\mu = \chi = \rho_0/c^2$ (3.33), we incorporate generic forces via

$$\partial_j \rho \to \partial_j \rho - \chi F_j^{\text{ext}}, \tag{3.43}$$

where, in the case of the Coulomb interaction, this "external" force takes the form

$$F_j^{\text{ext}}(\mathbf{x}, t) = -\partial_j \int \mathrm{d}^2\mathbf{x}'\, \frac{g}{|\mathbf{x} - \mathbf{x}'|}\,\rho(\mathbf{x}', t), \tag{3.44}$$

where $g$ is the Coulomb coupling in appropriate units; taking the Fourier transform of (3.44) gives

$$\widetilde{F}_j^{\text{ext}}(\mathbf{k}, \omega) = \int \mathrm{d}t\,\mathrm{d}^2\mathbf{x}\, e^{\mathrm{i}\omega t}\,e^{\mathrm{i}\mathbf{k}\cdot\mathbf{x}}\,F_j^{\text{ext}}(\mathbf{x}, t) = \mathrm{i}\,k_j\,\frac{2\pi g}{|\mathbf{k}|}\,\widetilde{\rho}(\mathbf{k}, \omega), \tag{3.45}$$

---

[5]In models with a circular Fermi surface, recent experimental work [45] has indeed aimed to detect the hydrodynamic crossover in plasmon dispersion.

and thus, we simply replace every instance of $\partial_j \rho = -\mathrm{i} k_j \widetilde{\rho}$ in (3.28) with

$$-\mathrm{i} k_j \widetilde{\rho} \rightarrow -\mathrm{i} k_j \left(1 + \frac{2\pi g \rho_0}{c^2 |\mathbf{k}|}\right) \widetilde{\rho}, \tag{3.46}$$

and we find the modified normal modes (with plasmons included) by solving (3.38) where the matrix $M$ (3.39) is replaced by the modified version,

$$M'(k_x, k_y) = \tag{3.47}$$

$$\begin{pmatrix} Dk(k + 2\pi\chi g) & \left(2\xi k_y - \mathrm{i}\right)k_x & -\mathrm{i} k_y + \xi\left(k_x^2 - k_y^2\right) \\ \left(2\xi k_y - \mathrm{i}\right)c^2 k_x \left(1 + \frac{2\pi g \chi}{k}\right) & \frac{1}{\rho_0}(\eta + \zeta)k_x^2 + \frac{1}{\rho_0}(\eta + \eta_\circ)k_y^2 & \frac{1}{\rho_0}(\zeta - \eta_\circ)k_x k_y \\ \left(\xi k_x^2 - \xi k_y^2 - \mathrm{i} k_y\right)c^2 \left(1 + \frac{2\pi g \chi}{k}\right) & \frac{1}{\rho_0}(\zeta - \eta_\circ)k_x k_y & \frac{1}{\rho_0}(\eta + \zeta)k_y^2 + \frac{1}{\rho_0}(\eta + \eta_\circ)k_x^2 \end{pmatrix},$$

where we used $\xi = \alpha/\rho_0 = \beta/c^2$ (3.35), $\chi = \rho_0/c^2$ (3.33), and the shorthand $k = |\mathbf{k}|$. Diagonalizing this matrix $M'$ to $O(k^2)$ gives rise to three quasinormal modes, one of which is the same "shear diffusion" mode reported in (3.41), while the other two "plasmon" (sound) modes are given by

$$\omega(\mathbf{k}) = \pm c \sqrt{2\pi\chi g k} \pm \frac{c^2 k^{3/2}}{2c\sqrt{2\pi\chi g}} - \mathrm{i}\pi D\chi g k - \frac{\mathrm{i}}{2\rho_0}\left(\rho_0 D + \eta + \zeta\right)k^2 \pm \mathrm{i}\xi c \sqrt{2\pi\chi g}\sin(3\theta)k^{3/2}, \tag{3.48}$$

where we have ignored a contribution to the second term proportional to $D^2 \sim \ell_{\mathrm{ee}}^2$. As before, stability requires that the imaginary part of $\omega$ be strictly negative, meaning $\pi g \chi D + \left(\rho_0 D + \eta + \zeta\right)k/2\rho_0 \geq c\xi\sqrt{2\pi g \chi k}$ (for $\theta = \pi/6$, e.g.). If $\xi$ saturates (3.36), then this condition becomes $k(D + \zeta/\rho_0)/2 + \left(\sqrt{\pi g \chi D} - \sqrt{\eta k/2\rho_0}\right)^2 \geq 0$ which is always satisfied for any values of the dissipative compatible with thermodynamic considerations.

In analogy to the plasmon-free case (3.41), the effect of the $D_6$ point group manifests in the decay rate $\propto \xi \sin(3\theta)k^{3/2}$ of the (modified) sound modes. As before, this new decay is proportional to $\xi$ and anisotropic in momentum space—the $\sin(3\theta)$ factor manifestly respects the threefold rotation symmetry. Note that inclusion of the Coulomb pressure will generically modify the sound modes above; however, we do not expect this to have a meaningful effect on either of the experiments we propose in Sec. 4.

## 3.5 Stream function

In the experimentally oriented section that follows, we will be interested in steady-state (i.e., time-independent) solutions of the equations of motion (3.28), in which case (3.28a) reduces to $\nabla \cdot j = 0$. Following previous work on the viscometry of materials with discrete point groups [17], we find it convenient to consider the "stream function" $\psi$[6] defined implicitly in terms of the current via

$$j_i = \rho_0 \epsilon_{ij} \partial_j \psi. \tag{3.49}$$

The continuity equations (3.28) can then be rewritten as

$$\rho_0 v_i = \rho_0 \epsilon_{ij} \partial_j \psi + D\partial_i \rho + \alpha\lambda_{ijk}\partial_j v_k, \tag{3.50a}$$

$$c^2 \partial_j \rho = \eta \nabla^2 v_j + \zeta\partial_j\left(\nabla \cdot v\right) - \eta_\circ \epsilon_{jk}\partial_k\left(\nabla \times v\right) + \beta\lambda_{jkl}\partial_k\partial_l\rho, \tag{3.50b}$$

where in (3.50a) we have replaced the density continuity equation (i.e., $\partial_i j_i = 0$) with the definition of the stream function, and the curl is given by $\nabla \times v = \partial_i \epsilon_{ij} v_j$.

---

[6]Note that the stream function $\psi$ is not well defined in a $2d$ domain with nontrivial first cohomology group. We do not consider such domains herein, and restrict to simply connected $2d$ geometries.

To leading order in $\ell_{ee}$, the equation of motion for the density $\rho$ is given by

$$\nabla^2 \rho = \frac{\beta}{c^2}\left(3\partial_x^2 - \partial_y^2\right)\partial_y \rho + O\left(\ell_{ee}^2\right), \tag{3.51}$$

where the LHS is $O(1)$ and the leading correction (on the RHS) is $O(\ell_{ee})$. The equation of motion for the stream function is then (at leading nontrivial order in $\xi$)

$$\nabla^4 \psi = \left(\frac{\alpha}{\rho_0} + \frac{\beta}{c^2}\right)\nabla^2\left(3\partial_x^2 - \partial_y^2\right)\partial_y \psi = 2\xi\nabla^2\left(3\partial_x^2 - \partial_y^2\right)\partial_y \psi = 2\xi\nabla^2\lambda_{ijk}\partial_i\partial_j\partial_k\psi, \tag{3.52}$$

where the RHS vanishes for $\xi = \alpha/\rho_0 = \beta/c^2 = 0$, recovering the standard biharmonic equation describing systems with continuous $O(2)$ rotational symmetry, or $D_{2M}$ dihedral symmetry, with $M \geq 4$ even. See App. A for a more detailed derivation of the equations of motion.

Solutions for the stream function are derived to $O(\xi)$ in App. A.7, and take the form

$$\psi(r,\theta) = \psi_0(r,\theta) + \xi\psi_1(r,\theta), \tag{3.53}$$

where $\psi_0$ is a solution to the biharmonic equation [46],

$$\nabla^4 \psi_0 = 0. \tag{3.54}$$

Solutions to (3.54) are given in polar coordinates in (A.31) [46]. We restrict to terms for which the current $j_i = \rho_0\,\epsilon_{ij}\partial_j\psi$ is nonsingular at the origin ($r \to 0$); we further dispense with terms corresponding to flows with fluid sources/sinks at $r = 0$ (i.e., $\psi \propto \theta \cdot f(r)$). The remaining solution to (3.54) then take the form

$$\psi_0(r,\theta) = a_0 r^2 + b_0 r^2 \ln r + \sum_{m=1}^{\infty}\left\{\left(a_m + b_m r^2\right)r^m\cos(m\theta) + \left(a'_m + b'_m r^2\right)r^m\sin(m\theta)\right\}, \tag{3.55}$$

and boundary conditions determine the values of the various coefficients.

The first correction $\psi_1$ at order $\xi$ satisfies

$$\nabla^4\psi_1 = 2\nabla^2\left(3\partial_x^2 - \partial_y^2\right)\partial_y\psi_0, \tag{3.56}$$

with $\psi_0$ given by (3.55). In App. B, we recover particular solutions for $\psi_1$ in complex coordinates; additionally, every allowed term in (3.55) is also allowed in $\psi_1$.

In polar coordinates, $\psi_1$ takes the general form

$$\psi_1(r,\theta) = b_0 r\sin(3\theta) + 12 b'_3 r^4 + \sum_{m=4}^{\infty}(m+1)m\,r^{m+1}\left\{b'_m\cos[(m-3)\theta] - b_m\sin[(m-3)\theta]\right\} + \dots, \tag{3.57}$$

where $\dots$ indicates that any solution to the biharmonic equation (3.54) is also allowed in $\psi_1$ (see App. B).

To leading order, the solution is given by

$$\begin{aligned}
\psi(r,\theta) = {}& \psi_0(r,\theta) + \xi\psi_1(r,\theta) + O(\xi^2)\\
= {}& a_0 r^2 + b_0 r^2\ln r + \sum_{m=1}^{\infty}\left\{\left(a_m + b_m r^2\right)r^m\cos(m\theta) + \left(a'_m + b'_m r^2\right)r^m\sin(m\theta)\right\}\\
& + 2\xi\tilde{a}_0 r^2 + 2\xi\tilde{b}_0 r^2\ln r\\
& + 2\xi\sum_{m=1}^{\infty}\left\{\left(\tilde{a}_m + \tilde{b}_m r^2\right)r^m\cos(m\theta) + \left(\tilde{a}'_m + \tilde{b}'_m r^2\right)r^m\sin(m\theta)\right\}\\
& + \xi b_0 r\sin(3\theta) + 12\xi b'_3 r^4\\
& + \xi r\sum_{m=4}^{\infty}(m+1)m\,r^m\left\{b'_m\cos[(m-3)\theta] - b_m\sin[(m-3)\theta]\right\} + O(\xi^2),
\end{aligned} \tag{3.58}$$

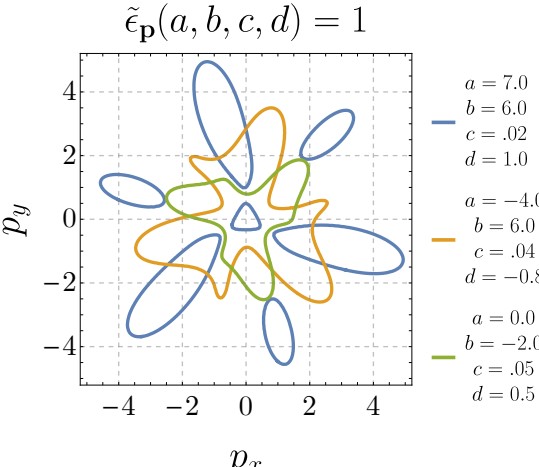

Figure 3: Contour plots of the $\mathbb{Z}_3$-invariant dispersion relation $\tilde{\varepsilon}$ (3.59). Fermi surfaces of this type possess only threefold discrete rotational symmetry. In particular, note that these Fermi surfaces possess no reflection symmetry axis.

where the various coefficients above are set by boundary conditions order by order in each order of $\xi$.

The effect of the $D_6$-symmetric terms is evident in the final line of (3.58), where we observe that the perturbation to the biharmonic equation (3.56) with coupling $\xi$ allows the angular harmonics (which, in an isotropic fluid, must be independent of one another by rotational symmetry) to mix according to $m \to m \pm 3$. We will see in Sec. 4 how this feature can be used to detect $\xi \neq 0$ unambiguously in experiments.

More formally, it is useful to note how the stream function $\psi$ can be broken up into the different irreps of the point group $D_6$. First note that in the O(2)-symmetric fluid, following [17], the $m = 0$ coefficients in (3.58) correspond to the $\mathcal{U}_0^-$ irrep, since $\psi$ is a pseudoscalar. The angular harmonics $\sin(m\theta)$ (and $\cos(m\theta)$) then combine and transform in the two-dimensional irrep $\mathcal{R}_m$. When O(2) is broken to $D_6$, any two irreps of O(2) that are equivalent upon restriction to $D_6$ might then be expected to mix due to the reduced symmetry of the hydrodynamic equations. Note in particular that the $R_1$ irrep of $D_6$ will contain every harmonic $m$ that is not a multiple of three, and by going to higher and higher orders in $\xi$, all of these harmonics can mix together through the last line of (3.58).

## 3.6 Breaking reflection invariance: $\mathbb{Z}_3$ fluids

Before discussing experimental proposals for detecting $\alpha$ and $\beta$, we turn briefly the consequences of breaking the reflection symmetry of $D_6$, which reduces $D_6$ to the cyclic group $\mathbb{Z}_3$, consisting solely of 120° rotations. A dispersion relation with $\mathbb{Z}_3$ symmetry is given, e.g., by

$$\tilde{\varepsilon}(\boldsymbol{p}) = a\left(p_x^2 + p_y^2\right) + b\left(3p_x^2 - p_y^2\right)p_y + c\left(p_x^2 + p_y^2\right)^4 + d\left(3p_x^5 p_y - 10p_x^3 p_y^3 + 3p_x p_y^5\right), \quad (3.59)$$

where $d \neq 0$ ensures the lack of reflection symmetry (e.g., under $p_x \to -p_x$; see also Fig 3).

A priori, this symmetry-breaking pattern could be rather dramatic: Now the point group is Abelian, and thus all irreps are one dimensional. The new invariant tensor under $\mathbb{Z}_3$ is $\epsilon_{ij}$ (which is parity odd and was therefore forbidden under $D_6$). However, one should not add terms containing $\epsilon_{ij}$ to the constitutive relations arbitrarily. For example, field-theoretic considerations suggest that one should not add a term $\tau_{ij} \sim \epsilon_{ij}\rho$ in a thermal system [34], and indeed, we find that such a term is not possible within our kinetic theory constructions

in Sec. 5. The momentum susceptibility must be a symmetric matrix, and so we cannot write down $\pi_i \sim a_1 v_i + a_2 \epsilon_{ij} v_j$. Consequently, we find that ideal hydrodynamics is *unchanged* from the $D_6$-invariant case, which itself was equivalent to isotropic ideal hydrodynamics.

One simple argument for this appears to be that time-reversal symmetry will, in general, include complex conjugation. The eigenvectors of the new $\mathbb{Z}_3$-invariant tensor $\epsilon_{ij}$ are of the form $v_x \pm i v_y$, corresponding to circularly polarized modes. Since these circularly polarized modes convert into one another under the action of time reversal, the combination of IT with this point group would form a non-Abelian symmetry group overall.

However, at first order in derivatives, we indeed find that new hydrodynamic coefficients are allowed. In particular, $\delta_{ij}$ and $\epsilon_{ij}$ are both $\mathbb{Z}_3$-invariant tensors; as a consequence, one can realize a viscosity tensor of the form

$$\eta^{\mathbb{Z}_3}_{ijkl} = \gamma \left( \delta_{ij}\epsilon_{kl} + \epsilon_{ij}\delta_{kl} \right), \tag{3.60}$$

where positivity requires that $\gamma^2 \leq \eta\,\eta_\circ$. We emphasize that this is *not* a Hall viscosity, which would couple to tensors of the form $\sigma^x_{ij}\sigma^z_{kl} - \sigma^z_{ij}\sigma^x_{kl}$ (which is antisymmetric upon exchanging the $ij$ and $kl$ indices). In fact, a Hall viscosity is not permitted in the $\mathbb{Z}_3$ fluid as long as IT symmetry is preserved.

The equations of motion (3.29) are then modified according to

$$\rho_0 \partial_t v_x = \dots - 2\gamma \partial_x \partial_y v_x + \gamma\left(\partial_x^2 - \partial_y^2\right) v_y, \tag{3.61a}$$

$$\rho_0 \partial_t v_y = \dots + \gamma\left(\partial_x^2 - \partial_y^2\right) v_x + 2\gamma \partial_x \partial_y v_y, \tag{3.61b}$$

where "..." indicates the terms that also appear in (3.28b). The modified quasinormal modes are given by a "sound mode," whose dispersion relation is

$$\omega_\pm(\mathbf{k}) = \pm c\,k - \frac{i}{2}\left(D + \eta + \zeta \mp 2\xi c \sin(3\theta)\right)k^2 \mp \left\{(D - \eta - \zeta)^2 - (2\gamma \pm 2\xi c \cos(3\theta))^2\right\} \frac{k^3}{8c}\cdots, \tag{3.62}$$

and a "shear diffusion mode" with

$$\omega_0(\mathbf{k}) = -i\left(\eta + \eta_\circ\right)k^2 + 2\xi\gamma \cos(3\theta)\,k^3\cdots, \tag{3.63}$$

where it is only at $O(k^3)$—which is subleading for all modes—that we see the effect of the new coefficient $\gamma$. The effect is most pronounced for the shear diffusion mode, which acquires a propagating and nondissipative contribution.

# 4 Experimental proposals

## 4.1 Hexagonal device

Here, we propose a class of experiments that can uniquely distinguish $D_6$ fluids in devices with symmetry-exploiting geometry. More specifically, following [17] we propose device geometries with specific boundary conditions on the current corresponding to particular irreducible representations of O(2) such that the current at the device center ($r \to 0$) is nonzero *only if* the fluid contained in the device has $D_6$ point group symmetry (or any subgroup thereof).

Regarding (3.58), the current at the center of a device with a $D_6$-invariant fluid subject to arbitrary boundary conditions has the following form for its Cartesian components:

$$j_x(r \to 0) = \rho_0 a_1 + \rho_0 \xi\left(\tilde{a}'_1 + b_0 \pm 2 b_0\right), \tag{4.64a}$$

$$j_y(r \to 0) = -\rho_0\left(a_1 + \xi \tilde{a}_1\right), \tag{4.64b}$$

where the $\pm$ above depends on whether $x \to 0$ is taken first (+) or $y \to 0$ is taken first (−). Hence for the current to be well defined everywhere, we must have $b_0 = 0$, and this will be the case for the boundary conditions of interest.

Using *group theoretic* principles, we can choose boundary conditions such that the current at the device center ($r \to 0$) is nonzero only if the fluid has point group $D_6$. Note that fluids with continuous rotational invariance (O(2)) or higher dihedral point group ($D_{2M}$ with $M > 3$) are governed by essentially the same continuity equations (3.28), but with $\xi = 0$. Thus, for the proposed device to distinguish $D_6$ rotational symmetry from other point groups, the current at the device center should vanish as $\xi \to 0$. Regarding (4.64b), this requires $a_1 = 0$, and to leading order in $\xi$, that one or both of $\widetilde{a}_1$ or $\widetilde{a}_1'$ are nonzero. By arranging leads as depicted in Fig. 4, we can guarantee this outcome from group theoretic principles alone.

With this choice of boundary conditions, only quantities compatible with said irreps can be nonzero at the device center. For fluids with higher rotational symmetry than $D_6$ (e.g. the symmetry group $D_{12}$ of a hexagon), the minimal such irrep is rank two; however, in general, such quantities cannot be measured directly (e.g., the stress tensor $\tau_{ij}$). Note that for a device with point group $D_6$, the $R_2$ irrep of $D_{12}$ is equivalent to the $R_1$ irrep of $D_6$, which admits a nonzero *rank-one* signal, realized by a nonzero current at the device center. Hence, by affixing current-carrying leads to the sample in a particular pattern, one can realize boundary conditions that force the current to vanish at the device center *unless* the device has point group $D_6$ (or a subgroup thereof). Fig. 4 depicts a particular arrangement of current-carrying leads on a sample that will lead to a nonzero current at the device center only if the point group is $D_6$ (or a subgroup thereof). For fluids with continuous (or discrete but morefold) rotational invariance, only a rank-two signal can be nonzero at the center; in general, such quantities cannot be readily measured. Hence, only fluids with point group $D_6$ (or any of its subgroups) allow for a nonzero current at the center.

More precisely, the hexagon boundary conditions in Fig. 4 transform under the $D_{12}$ irrep $S = R_2$, and the circle boundary conditions in Fig. 4 transform under the O(2) representation $S = \bigoplus_{m \in \mathbb{N} \backslash 3\mathbb{N}} \mathcal{R}_{2m}$. In either case, $G = D_6$ is the *only* orthogonal subgroup $G \leq O(2)$ for which the $G$-restricted boundary representation $S|_G$ contains the *vector* (spin-one) irrep $R_1$. Since the center of the hexagon (or circle) is a fixed point of $D_{12} / O(2)$, this implies that a nonzero vector is allowed at the center of either device *only* when the contained fluid has point group $G = D_6$. Observation of a nonzero current (4.66) at the center of either device can therefore uniquely distinguish $D_6$ fluids.

For concreteness and analytical convenience, we consider the right panel of Fig. 4, corresponding to a circular device, and assume the leads to be infinitesimally thin (i.e., modeled by delta functions of the polar angle). However, any arrangement of leads corresponding to the two-dimensional irrep $R_2$ of $D_{12}$ (or equivalently, irreps $\mathcal{R}_k$ of O(2) with $k$ even but not a multiple of three), will correctly distinguish $D_6$ fluids from those with other point groups.

A detailed derivation of the current flow $j_i$, to first order in $\xi$, can be found in App. B; we present only the main results here. In App. B.1 we impose boundary conditions on the current that derives from $\psi$ according to (3.49); these boundaries correspond to attaching radially oriented and infinitesimally thin wires to the sample's edge in a pattern corresponding to the right panel of Fig. 4. In App. B.2 we find the resulting current in the sample bulk in radial coordinates, and in App. B.3, we find the corresponding stream function.

The current at the device center is most easily evaluated by regarding the stream function in Cartesian coordinates—to lowest order in $x$ and $y$, the stream function is given by

$$\psi(x, y) = \frac{j_{\text{in}} \sqrt{3}}{\pi \rho_0 R} \left( \frac{x^4 - y^4}{R^2} + y^2 - x^2 - 20\,\xi\,R^2\,y + 40\,\xi\,\frac{x^2 + y^2}{R^2} \right) + O(r^5), \qquad (4.65)$$

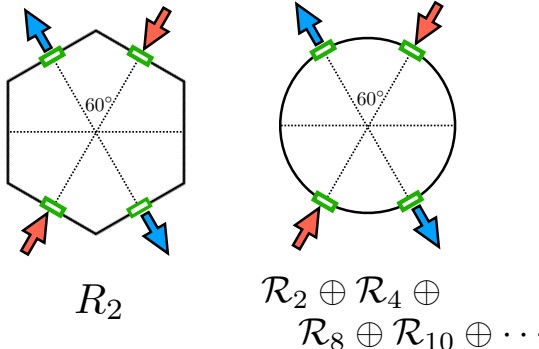

$R_2$

$\mathcal{R}_2 \oplus \mathcal{R}_4 \oplus$
$\mathcal{R}_8 \oplus \mathcal{R}_{10} \oplus \cdots$

Figure 4: Hexagonal (left) and circular (right) devices with symmetry-engineered boundary conditions. The hexagonal device is likely more feasible for experiment, but for analytical convenience we will assume boundary conditions of the circular type. Leads (green) are placed on the boundary, with current injected (red) or drained (blue) orthogonally at each lead. Due to symmetry considerations, a nonzero current can appear at the center of either device *only* if the contained fluid has $D_6$ point symmetry, thus providing a unique experimental signature of $D_6$ fluids.

and the resulting current at the origin is given in Cartesian coordinates by

$$j_x = -\frac{20\sqrt{3}}{\pi}\frac{\xi}{R}j_{\text{in}}, \quad j_y = 0, \tag{4.66}$$

which is proportional to the current through leads times the ratio $\ell_{\text{ee}}/R$; the coefficient of proportionality is $\sim 10$.

As a reminder, while the exact value of the current at the device center (4.66) assumes a particular arrangement of infinitesimally thin current-carrying leads, any arrangement of leads corresponding to the $R_2$ irrep of $D_{12}$ will result in a nonzero signal of order $\xi/R$ times the current through the leads. This can be understood in terms of the harmonic expansion of the stream function (3.58).

In a previous study of inversion-symmetric fluids [17], numerous quantities could be detected and isolated in measurements of *heating* at the center of a device with appropriate geometry and arrangement of fluids. This conveniently provides for the isolation of various dissipative coefficients in fluids with $D_{2M}$ symmetry with $M = 4, 6$ and higher. However, we note that this is not possible in the $D_6$ fluid, as the contribution to heating from $\xi \neq 0$ cannot be isolated from contributions due to other terms (i.e., the shear viscosity $\eta$, and incoherent diffusion $D$). Hence, arranging leads on a sample as depicted in Fig. 4 and detecting a nonzero current at the device center is the only way to isolate the effects of $\xi \neq 0$ using the viscometric principles of [17].

Lastly, we remark that there is another arrangement of leads, corresponding roughly to a 90° rotation of the arrangement shown in Fig. 4, which would also lead to $R_2$ boundary conditions. The effect one would observe is equivalent to what we write above, except with $j_x = 0$ and $j_y \neq 0$.

## 4.2 Hall effect in narrow channels

Now we argue that, in contrast to the hexagonal device experiment presented in Sec. 4.1 (which records a clear signal at the lowest possible order in $\xi$), it is generally quite difficult to detect $\xi \neq 0$ (i.e. $D_6$ point group symmetry) in a more standard Hall transport experiment in an electronic system. The goal is to see a "Hall voltage", realized in this case by a potential

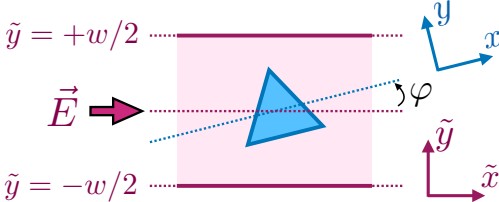

Figure 5: Electron fluid with $D_6$ Fermi surface (blue) forced down a narrow channel (purple) by an applied electric field $\vec{E}$. $D_6$ point anisotropy of the contained fluid will generically lead to a $\varphi$-dependent Hall effect signal, i.e. a voltage difference between the channel walls $\tilde{y} = \pm w/2$; see Eq. (4.76).

difference across the channel in response to an electric field oriented along the channel. We imagine cutting the $D_6$ sample into a narrow channel of width $w$, whose "walls" make an angle $\varphi$ with the "original" (crystallographic) $x$ axis. In the case where the Fermi surface is a regular triangle, (3.28) assumes one of the edges of the triangle to be aligned with the $x$ axis. Thus, $\varphi$ is also the angle between the "base" of the Fermi surface and the channel walls, as depicted in Fig. 5.

Naively, one might expect that such an experiment is quite effective for detecting the symmetry breaking pattern of the $D_6$ fluid. For example, if the channel is carved at a generic angle relative to the triangular Fermi surface, the combination of the channel and the device completely breaks the rotational symmetry of the device, and so we expect a Hall voltage $V_y = R_{yx} I$, with $R_{yx} \neq 0$ as it is no longer forbidden by any symmetries.

However, we note that (*i*) $R_{yx}$ is quite sensitive to boundary conditions; (*ii*) even assuming one did perfectly know the boundary conditions in an experimental setup, $R_{yx}$ can appear at order $\xi^2$ which is a subleading signal relative to that in the hexagonal device previously discussed; (*iii*) most importantly, this Hall effect is not unique to $D_6$ fluids. For these reasons, we do not suggest using Hall transport as a probe for novel transport phenomena in triangular electron fluids. We now justify these conclusions in more detail.

The only terms in the continuity equations that are not invariant under rotation of the coordinate axes are those with coefficient $\xi$, which arise from the $D_6$ rotational symmetry of the Fermi surface, and are thus sensitive to its orientation. A derivation of these terms in the rotated coordinates $(\tilde{x}, \tilde{y})$ is given in App. C.2. We then seek steady-state "flow" solutions that do not vary with time, nor the distance $\tilde{x}$ along the channel.

We also modify (3.28) by applying an electric field along the $\tilde{x}$ axis,[7] and look for steady-state solutions (i.e., take $\partial_t \to 0$) that do not vary in the $\tilde{x}$ direction (i.e., the distance along the channel). The desired signal—a Hall voltage—is indicated by a nonzero difference in density $\rho$ between the two walls, i.e.,

$$V_{\text{Hall}} \propto \rho(\tilde{y} = w/2) - \rho(\tilde{y} = -w/2), \tag{4.67}$$

which we expect to be zero—at least in the context of first-order hydrodynamics—for fluids with continuous rotational invariance or higher dihedral point group $D_{2M}$ (with $M \geq 4$). For point group $D_6$, a nonzero Hall voltage is allowed by symmetry; as we will show, we generally expect a signal proportional to $\xi^2$.

---

[7]This can be viewed as applying an external chemical potential: $\mu \to \mu - E\tilde{x}$.

### 4.2.1 Summary of results

The equations of motion for Poiseuille flow are given by

$$\rho_0 \partial_{\tilde{y}} v_{\tilde{y}} = D\,\partial_{\tilde{y}}^2 \rho - \alpha\,\sin(3\varphi)\,\partial_{\tilde{y}}^2 v_{\tilde{x}} - \alpha\,\cos(3\varphi)\,\partial_{\tilde{y}}^2 v_{\tilde{y}}\,, \tag{4.68a}$$

$$-c^2\,\chi\,E = \left(\eta + \eta_\circ\right)\partial_{\tilde{y}}^2 v_{\tilde{x}} - \beta\,\sin(3\varphi)\,\partial_{\tilde{y}}^2 \rho\,, \tag{4.68b}$$

$$c^2 \partial_{\tilde{y}} \rho = \left(\eta + \zeta\right)\partial_{\tilde{y}}^2 v_{\tilde{y}} - \beta\,\cos(3\varphi)\,\partial_{\tilde{y}}^2 \rho\,, \tag{4.68c}$$

where $E$ is the strength of the electric field (oriented along the channel in the $\tilde{x}$ direction), and we restrict to solutions that do not depend on the time $t$ nor the distance $\tilde{x}$ along the channel. The components of the current are then given by

$$j_{\tilde{x}} = \rho_0 v_{\tilde{x}} + D\,\chi\,E - \alpha\,\cos(3\varphi)\,\partial_{\tilde{y}} v_{\tilde{x}} + \alpha\,\sin(3\varphi)\,\partial_{\tilde{y}} v_{\tilde{y}}\,, \tag{4.69a}$$

$$j_{\tilde{y}} = \rho_0 v_{\tilde{y}} - D\,\partial_{\tilde{y}} \rho + \alpha\,\sin(3\varphi)\,\partial_{\tilde{y}} v_{\tilde{x}} + \alpha\,\sin(3\varphi)\,\partial_{\tilde{y}} v_{\tilde{y}}\,, \tag{4.69b}$$

and we now seek solutions to the above equations with sensible boundary conditions.

To obtain a more convenient form of the solution, we integrate (4.68c) once and take the constant of integration to be zero; this particular choice does not have an effect on the Hall voltage, nor the solutions themselves (allowing the constant of integration to be arbitrary simply amounts to a constant shift in $\rho$, which does not affect the Hall voltage, currents, or equations of motion). However, this choice greatly simplifies matching boundary conditions.

The equations of motion (4.68) can be solved exactly, with solutions given by

$$\rho(\tilde{y}) = \xi\,\chi\,E\,\sin(3\varphi)\,\frac{\eta + \zeta}{\eta + \eta_\circ} + C_+\,e^{-\tilde{y}/\ell_+} + C_-\,e^{-\tilde{y}/\ell_-}\,, \tag{4.70a}$$

$$v_{\tilde{x}}(\tilde{y}) = A_0 + A_1\,\tilde{y} - \frac{\rho_0 E}{\eta + \eta_\circ}\,\frac{\tilde{y}^2}{2} + \frac{\xi\,c^2}{\eta + \eta_\circ}\,\sin(3\varphi)\left[C_+\,e^{-\tilde{y}/\ell_+} + C_-\,e^{-\tilde{y}/\ell_-}\right]\,, \tag{4.70b}$$

$$v_{\tilde{y}}(\tilde{y}) = B_0 + \frac{\rho_0 \xi E \sin(3\varphi)}{\eta + \eta_\circ}\,\tilde{y} + \frac{c^2}{\eta + \zeta}\left[\left(\xi \cos(3\varphi) - \ell_+\right)C_+\,e^{-\tilde{y}/\ell_+} + \left(\xi \cos(3\varphi) - \ell_-\right)C_-\,e^{-\tilde{y}/\ell_-}\right]\,, \tag{4.70c}$$

where $A_{0,1}$, $B_0$, and $C_\pm$ are constants of integration, to be determined by boundary conditions, and the two lengthscales $\ell_\pm$ can be written in terms of hydrodynamic coefficients and the misalignment $\varphi$ as

$$\ell_\pm = \frac{\rho_0 c^2 \xi^2 \left(\eta + \eta_\circ \cos^2(3\varphi) + \zeta \sin^2(3\varphi)\right) - D\left(\eta + \eta_\circ\right)(\eta + \zeta)}{\rho_0 c^2 \xi \left(\eta + \eta_\circ\right)\cos(3\varphi) \mp \sqrt{\rho_0 c^2 \left(\eta + \eta_\circ\right)(\eta + \zeta)\left(D\left(\eta + \eta_\circ\right) - \rho_0 c^2 \xi^2 \sin^2(3\varphi)\right)}}\,, \tag{4.71}$$

and are both proportional to $\ell_{\mathrm{ee}}$.

The general result for the Hall voltage is given by

$$V_{\mathrm{Hall}} \propto \rho(\tilde{y})\big|_{-w/2}^{w/2} = 2\sum_\pm C_\pm \sinh\left[w/2\ell_\pm\right]\,, \tag{4.72}$$

which we note is sensitive to boundary conditions, to which we now turn.

Regarding the solutions (4.70), it is difficult to fix physical boundary conditions for arbitrary $\xi$. To make progress—and to determine at what order in $\xi$ a Hall voltage appears—we expand the solutions order by order in $\xi$, allowing the coefficients $A_{0,1}$, $B_0$, and $C_\pm$ arbitrary dependence on $\xi$. We set $j_{\tilde{y}}$ to zero, and demand that $v_{\tilde{y}}$ vanish at the channel walls, as these choices are both physically sensible and sufficient for the Hall voltage to vanish when $\xi = 0$, which we expect on general symmetry grounds. Additionally, we require that $v_{\tilde{x}}$ be symmetric at the two walls, i.e. $v_{\tilde{x}}(w/2) = v_{\tilde{x}}(-w/2)$, where $w$ is the channel width.

This is a plausible choice of boundary conditions, but not required by any symmetry principle; however, we note that (*i*) these boundary conditions reproduce the standards results for fluids with $O(2)$ symmetry and (*ii*) these choices are *least* favorable to seeing a nonzero Hall voltage at low order in $\xi$. Hence, we expect our results reflect the minimal signal one can expect in such an experiment, are entirely consistent with typical boundary conditions, and reproduce the expected results as $\xi \to 0$ [16, 47].

Before proceeding to the expansion, we note that this choice of boundary conditions is *least* favorable to a nonzero Hall voltage: The signal resulting from this choice is the minimal result that can be expected in a real experiment; more exotic choices of boundary conditions may lead to a stronger Hall voltage (i.e., one that appears at lower order in the small parameter $\xi \propto \ell_{\text{ee}}$).

Imposing these boundary conditions, we find

$$A_0 = \frac{D\,E}{c^2}\frac{\eta+\zeta}{\eta+\eta_\circ}\frac{\tilde{w}^2}{2} + \frac{\rho_0\,E\,(\eta+\zeta)\sin^2(3\varphi)}{\left(\eta+\eta_\circ\right)^2}\,\tilde{w}\coth(\tilde{w})\,\xi^2 + O(\xi^3), \tag{4.73a}$$

$$A_1 = O(\xi^3), \tag{4.73b}$$

$$B_0 = -\frac{\rho_0\,E\,\sin(6\varphi)}{2\left(\eta+\eta_\circ\right)}\xi^2 + O(\xi^3), \tag{4.73c}$$

$$C_\pm = -\frac{1}{2}\chi\,E\,\sin(3\varphi)\frac{\eta+\zeta}{\eta+\eta_\circ}\operatorname{csch}(\tilde{w})\,\tilde{w}\,\xi \mp \frac{\chi\,E\,\sin(6\varphi)}{2\,\ell_0}\frac{\eta+\zeta}{\eta+\eta_\circ}\operatorname{sech}(\tilde{w})\left(1+\tilde{w}^2\right)\xi^2 + O(\xi^3), \tag{4.73d}$$

where we make use of the following two length scales,

$$\ell_0 = \lim_{\xi\to 0}\ell_\pm = \sqrt{\frac{D\,(\eta+\zeta)}{\rho_0\,c^2}}, \quad \text{and} \quad \tilde{w} = \frac{w}{2\,\ell_0} = \frac{w}{2}\sqrt{\frac{\rho_0\,c^2}{D\,(\eta+\zeta)}}, \tag{4.74}$$

which are, respectively, another length scale proportional to $\ell_{\text{ee}}$ and a dimensionless parameter proportional to $w/\ell_{\text{ee}}$.

Using these results, the Hall voltage is

$$V_{\text{Hall}} \propto \Delta\rho = \xi^2\,\chi\,E\,\sin(6\varphi)\frac{\eta+\zeta}{\eta+\eta_\circ}\frac{1}{\ell_0}\tanh(\tilde{w})\left\{1+\tilde{w}^2\left(1-\coth^2(\tilde{w})\right)\right\}+O(\xi^3), \tag{4.75}$$

which we can simplify by noting that $\tanh(\tilde{w})$ and $\coth(\tilde{w})$ approach unity for $\tilde{w}\gg 1$. In fact, this corresponds to the limit of interest $w\gg\ell_{\text{ee}}$: So long as the channel width is noticeably larger than the electron-electron scattering length $\ell_{\text{ee}}$ we can safely take $\coth(\tilde{w})=\tanh(\tilde{w})=1$ in (4.75), to recover

$$V_{\text{Hall}} \propto \xi^2\,\chi\,E\,\sin(6\varphi)\frac{\eta+\zeta}{\eta+\eta_\circ}\frac{1}{\ell_0} = \frac{\alpha\beta}{c^4}E\,\sin(6\varphi)\frac{\eta+\zeta}{\eta+\eta_\circ}\sqrt{\frac{\rho_0\,c^2}{D\,(\eta+\zeta)}}, \tag{4.76}$$

which is proportional to $\ell_{\text{ee}}$ overall, much like the current signal in the hexagonal device experiment. However, the Hall voltage here is proportional to $\xi^2$, and thus higher order in the $D_6$ coefficient than the current at the center of the hexagon, which is $O(\xi)$.

For comparison, taking $A_0 \to 0$ so that there is no constant contribution to the velocity $v_{\tilde{x}}$ along the channel, the longitudinal conductivity can be extracted from Ohm's law according to

$$\sigma_{\tilde{x}\tilde{x}} = \frac{1}{E\,w}\int_{-w/2}^{w/2}\mathrm{d}\tilde{y}\,j_{\tilde{x}} = \chi\,D + \frac{\rho_0^2\,w^2}{12\left(\eta+\eta_\circ\right)} + O(\xi^2), \tag{4.77}$$

where we have averaged the conductivity across the channel (i.e., the $\tilde{y}$ direction).

From (4.76) we see that $V_{\text{Hall}} \sim \xi^2/\ell_{\text{ee}}$. In contrast, using (4.77), the longitudinal voltage will roughly scale as $V_{\text{long}} \sim (\ell_{\text{ee}} + w^2/\ell_{\text{ee}})L/w$ in a channel of length $L$, which is significantly larger than $V_{\text{Hall}}$ in the hydrodynamic regime (since $\xi \ll w$), even when correcting for the overall geometric prefactor of $L/w$. Thus the Hall voltage signal is rather weak, which can be quantified by noting that the "Hall angle" $\theta_{\text{H}}$ which measures the relative angle between the current $j$ and electric field $E$ will scale as $\theta_{\text{H}} \sim (\xi/w)^2 \ll 1$.

A curious feature of (4.76) is the presence of sixfold—rather than threefold—rotational symmetry. This is a consequence of the fact that the signal is proportional to $\xi^2$, rather than $\xi$. In order to see a signal with threefold rotational symmetry, one must devise (or realize) boundary conditions that are more sensitive to $\xi$.

Hence, we generally expect a nonzero Hall voltage across narrow channels with $D_6$-invariant fluids under a longitudinal electric field. At the same time, we also expect that this signal can be weaker than the current in the hexagonal device experiment proposed in Sec. 4.1. We also note that it may be possible to observe a stronger signal by enforcing other boundary conditions, but (*i*) the prediction (4.76) constitutes the minimal signal one can expect, and (*ii*) there is not a microscopic determination of boundary conditions (besides trial and error) in an actual experiment. This means that the Hall signal may be a rather poor test for $\alpha, \beta \neq 0$ since the experimental signal is too sensitive to model details. We have also confirmed explicitly that including momentum relaxation processes in (3.28) does not qualitatively change the resulting Hall voltage (4.76)—because the corresponding formulae are uninspiring to behold, we have not included them here.

### 4.2.2 Nonuniqueness of the Hall voltage signal

Another reason we claim that the Hall effect signal is inferior to the "hexagonal device" as a probe of $\alpha$—despite its relative simplicity to realize experimentally—is that the Hall effect in the channel is not unique to the inversion-broken fluid. In other words, it is not the breaking of inversion symmetry that is responsible for a Hall voltage signal, since $\mathcal{I} \cdot \sigma_{xy} = \sigma_{xy}$ is already invariant. What blocks a Hall voltage from appearing in experiment is the presence of $y \to -y$ parity symmetry. Regardless of the Fermi surface shape, as long as it is not a circle, this symmetry can be broken by orienting the Fermi surface at a sufficiently generic angle with the channel.

Fig. 6 shows how in a (toy model of an) inversion-symmetric ($D_{12}$-invariant) Fermi liquid, in the ballistic limit, it is possible to find a nonvanishing Hall voltage when the Fermi surface is rotated relative to the channel walls. The kinetic theory model of transport (including boundary conditions on the channel) is described in [16]; since we used this model exactly as written in [16], we will not reproduce the technical details here. We also note that a similar effect has been seen in a recent experiment on $PdCoO_2$ [37]; note that $PdCoO_2$ has an approximately hexagonal Fermi surface, and the experiment was done in a more complicated geometry.

## 5 Kinetic theory

In this section, we develop a low-temperature kinetic theory for $D_6$ fluids. In particular, we study the linear response regime of the Boltzmann equation for the quasiparticle distribution function $f(x, p)$ assuming that charge and momentum are the only relevant conserved quantities. The Boltzmann equation for $f$ will depend on microscopic, band-theoretic details (e.g., the dispersion relation $\epsilon_p$ and the linearized collision operator W), which we model phenomenologically, subject to the restrictions required by demanding $D_6$ invariance.

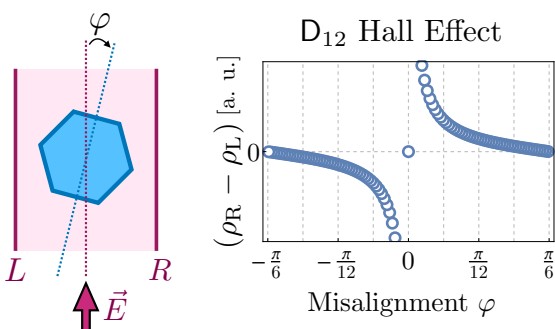

Figure 6: The presence of a Hall voltage signal is not unique to inversion-symmetry breaking $D_6$ fluids. *Left*: Electron fluid with $D_{12}$ Fermi surface (blue) forced down a narrow channel (purple) by an applied electric field $\vec{E}$. Note that $D_{12}$ possesses inversion $p \rightarrow -p$ symmetry, in contrast to $D_6$. *Right*: Misalignment $\varphi$ between the Fermi surface and channel creates a particle density difference $\Delta\rho = (\rho_R - \rho_L)$, and hence Hall voltage $V_H \propto \Delta\rho$, between the channel walls. The density difference $\Delta\rho(\varphi)$, plotted here in arbitrary units, is numerically obtained from the kinetic theory model developed in [16], with absorbing boundary conditions at the channel walls. Note that the divergence of $\Delta\rho(\varphi \rightarrow 0)$ in this model [16] reflects the assumption of perfect momentum conservation in the bulk; in a physical system, this divergence would be instead regulated by a nonzero rate $\Gamma$ for momentum relaxation (e.g. impurity, Umklapp scattering).

## 5.1 Formalism

We begin by describing the kinetic theory formalism following [48]. First, we expand the distribution function $f$ around the Fermi-Dirac distribution,

$$f_{\text{eq}}(p) = \left[1 + e^{\beta(\epsilon_p - \mu)}\right]^{-1}, \tag{5.78}$$

evaluated at equilibrium:

$$f(x, p) = f_{\text{eq}}(p) + \left(-\frac{\partial f_{\text{eq}}}{\partial \epsilon_p}\right)\Phi(x, p) + O\left(\Phi^2\right). \tag{5.79}$$

We focus on the distribution perturbation $\Phi$ in the linear response regime (i.e., higher powers of $\Phi$ are neglected). Since

$$\left(-\frac{\partial f_{\text{eq}}}{\partial \epsilon}\right) = \delta(\epsilon - \mu) + O\left(\frac{k_B T}{\mu}\right), \tag{5.80}$$

extracts the low-temperature singularities in $f - f_{\text{eq}}$, and the distribution perturbation $\Phi$ is typically well-behaved as $T \rightarrow 0$; we therefore focus on the dynamics of $\Phi$. Following the linear algebra formalism detailed in [48], we then introduce the distribution vector

$$|\Phi(x)\rangle = \int d^2 p \, \Phi(x, p) |p\rangle, \tag{5.81}$$

integrated over a basis of momentum kets $|p\rangle$ subject to the inner product

$$\langle p | p' \rangle = \left(-\frac{\partial f_{\text{eq}}}{\partial \epsilon_p}\right)\frac{\delta(p - p')}{(2\pi\hbar)^2}. \tag{5.82}$$

Note that the factor $\left(-\partial f_{\mathrm{eq}}/\partial \epsilon\right)$ (5.80) reduces inner products of distribution kets $|\Phi\rangle$ (5.81) to integrals along the Fermi surface (in the low-temperature limit).

Taking the Fourier transform $\nabla \to ik$, the (source-free) linearized Boltzmann equation for $|\Phi\rangle$ becomes [48]

$$(\partial_t + \mathsf{W} + \mathsf{L}) |\Phi(k)\rangle = 0 \,, \tag{5.83}$$

where

$$\mathsf{L} = v(p) \cdot \nabla \to ik \cdot v(p) = ik \cdot (\partial_p \epsilon_p) \,, \tag{5.84}$$

is the streaming operator, which models the convective transport of distribution kets, and $\mathsf{W}$ is the linearized collision integral, which models the decay of (nonconserved) kets with time.

Note that $\mathsf{W}$ must be positive semi-definite, so as to exclude the possibility of negative-eigenvalue eigenmodes of $\mathsf{W}$ exponentially growing in time via Eq. (5.83). Note also that $\mathsf{W}$ must be symmetric as a consequence of IT symmetry [16]. Indeed, under the two symmetries, momentum kets transform as

$$\mathcal{I} |p\rangle = |-p\rangle \,, \tag{5.85a}$$

$$\Theta |p\rangle = |-p\rangle \,, \tag{5.85b}$$

while the collision integral must transform, e.g., as

$$\Theta \cdot \left\langle p_1 \middle| W \middle| p_2 \right\rangle = \left\langle \Theta p_2 \middle| W \middle| \Theta p_1 \right\rangle \,, \tag{5.86}$$

and combining these two identities, we conclude that IT symmetry requires a symmetric collision integral:

$$\left\langle p_1 \middle| \mathsf{W} \middle| p_2 \right\rangle = \left\langle p_2 \middle| \mathsf{W} \middle| p_1 \right\rangle \,. \tag{5.87}$$

Hydrodynamics emerges in the long-time and long-wavelength limit of the kinetic theory; this limit is taken by "integrating out" the nonconserved modes (i.e., eigenmodes of $\mathsf{W}$ with positive eigenvalue) in the equation of motion (5.83), leaving only the conserved modes as dynamical modes.

We now outline this "integrating out" procedure. Consider a solution $|\Phi\rangle$ of the source-free Boltzmann equation (5.83). Letting $a$ label the dynamical conserved modes (i.e. eigenmodes of $\mathsf{W}$ with zero eigenvalue) and $b$ label the decaying modes to be integrated out (i.e. eigenmodes of $\mathsf{W}$ with positive eigenvalue), we write the Boltzmann equation (5.83) in a block-diagonal basis of $\mathsf{W}$ as

$$\left[ \partial_t + \begin{pmatrix} 0 & 0 \\ 0 & \mathsf{W}_b \end{pmatrix} + \begin{pmatrix} \mathsf{L}_a & \mathsf{L}_{ab} \\ -\mathsf{L}_{ab}^\dagger & \mathsf{L}_b \end{pmatrix} \right] \left( \begin{vmatrix} \Phi_a \\ \Phi_b \end{vmatrix} \right) = \begin{pmatrix} 0 \\ 0 \end{pmatrix} \,, \tag{5.88}$$

where we have used the fact that the streaming operator $\mathsf{L}$ is anti-Hermitian (after Fourier transforming $\nabla \to ik$). In the hydrodynamic limit (i.e. on timescales for which the $b$ modes have effectively decayed away) we have that $\partial_t \ll \mathsf{W}_b$. Hence, in this limit it is reasonable to approximate $\partial_t \approx 0$ in the $b$-sector equation. We then solve the $b$-sector equation for the modes $|\Phi_b\rangle$ and substitute the result into the $a$-sector equation, giving

$$\left[ \partial_t + \mathsf{L}_a + \mathsf{L}_{ab} \left( \mathsf{W}_b + \mathsf{L}_b \right)^{-1} \mathsf{L}_{ab}^\dagger \right] |\Phi_a\rangle = 0 \,. \tag{5.89}$$

Finally, since $\mathsf{W}_b \sim \gamma_b$ and $\mathsf{L}_b \sim v_{\mathrm{F}} k$, we further have that $\mathsf{L}_b \ll \mathsf{W}_b$ in the hydrodynamic limit. Thus we may to good approximation take $\left( \mathsf{W}_b + \mathsf{L}_b \right)^{-1} \approx \mathsf{W}_b^{-1}$, so that

$$\left( \partial_t + \mathsf{W}' + \mathsf{L}_a \right) |\Phi_a\rangle = 0 \tag{5.90}$$

in the hydrodynamic regime, with the effective collision integral

$$\mathsf{W}' = \mathsf{L}_{ab}\,\mathsf{W}_b^{-1}\,\mathsf{L}_{ab}^{\dagger}\,, \tag{5.91}$$

which is the origin of diffusive contributions to the hydrodynamic equations for the conserved modes $\left|\Phi_a\right\rangle$.

The modifications of the low-temperature distribution functions that are compatible with the conservation of charge and momentum are of the form

$$\left|\Phi\right\rangle = \delta\mu\left|\rho\right\rangle + \delta v_i\left|p_i\right\rangle\,, \tag{5.92}$$

where we have defined

$$\left|\rho\right\rangle = \int \mathrm{d}^2 p\,\left|p\right\rangle\,, \tag{5.93a}$$

$$\left|p_i\right\rangle = \int \mathrm{d}^2 p\,p_i\left|p\right\rangle\,, \tag{5.93b}$$

and these three modes are the slow $a$ modes from the discussion above. Inserting (5.92) into (5.83), we find

$$\partial_t\left\langle\rho|\Phi\right\rangle + \mathrm{i}\,k_i\left\langle j_i|\Phi\right\rangle = 0\,, \tag{5.94a}$$

$$\partial_t\left\langle p_j\middle|\Phi\right\rangle + \mathrm{i}\,k_i\left\langle \tau_{ij}\middle|\Phi\right\rangle = 0\,, \tag{5.94b}$$

where

$$\left|j_i\right\rangle = \int \mathrm{d}^2 p\,v_i(p)\left|p\right\rangle\,, \tag{5.95a}$$

$$\left|\tau_{ij}\right\rangle = \int \mathrm{d}^2 p\,v_i(p)\,p_j\left|p\right\rangle \tag{5.95b}$$

represent the projections onto the charge current and stress tensor (momentum current). Expressions for $\left|j\right\rangle$ and $\left|\tau\right\rangle$ come from multiplying $\mathsf{L}$ onto the conserved modes. The projection of $\left|j\right\rangle$ and $\left|\tau\right\rangle$ onto $\left|\rho\right\rangle$ and $\left|p\right\rangle$ will lead to ideal hydrodynamics (the $\mathsf{L}_a$) terms in (5.90)). First-order hydrodynamics arises from the $\mathsf{W}'$ terms in (5.90); the next two subsections detail each case in turn.

## 5.2 Zeroth-order hydrodynamics

To obtain zeroth-order (ideal) hydrodynamics, we must project the currents onto the density modes. This means that we approximate

$$\left\langle j_i|\Phi\right\rangle \approx \left\langle j_i\middle|\left(\delta\mu\left|\rho\right\rangle + \delta v_j\middle|p_j\right\rangle\right)\,. \tag{5.96}$$

Now, the $\mathsf{D}_6$ symmetry group immediately implies that

$$\left\langle j_i|\rho\right\rangle = 0\,, \tag{5.97}$$

as these two objects transform in different irreps. Hence we must calculate instead

$$\begin{aligned}
\left\langle j_i\middle|p_j\right\rangle &= \int \mathrm{d}^2 p\left(-\frac{\partial f_{\mathrm{eq}}}{\partial\epsilon}\right)v_i\,p_j = \int \mathrm{d}^2 p\left(-\frac{\partial f_{\mathrm{eq}}}{\partial\epsilon}\right)\frac{\partial\epsilon}{\partial p_i}p_j \\
&= \int \mathrm{d}^2 p\left(-\frac{\partial f_{\mathrm{eq}}}{\partial p_i}\right)p_j = \delta_{ij}\int \mathrm{d}^2 p\,f_{\mathrm{eq}} = \rho_0\delta_{ij}\,.
\end{aligned} \tag{5.98}$$

Therefore within ideal hydrodynamics,

$$j_i \approx \rho_0 \delta v_i, \tag{5.99}$$

in agreement with our earlier results in Sec. 3. A similar calculation reveals that

$$\left\langle \tau_{ij} \middle| \rho \right\rangle = \rho_0 \delta_{ij}, \tag{5.100}$$

which can be understood by noting that the density can only overlap with the $U_0^+$-component of $\tau$, but holds even more generally on thermodynamic grounds [34]. Interestingly, we observe that $\tau_{ij}$ does have an $R_1$ component, so let us check whether or not $\tau_{ij}$ could have a coefficient proportional to $\delta v_i$:

$$\left\langle \tau_{ij} \middle| p_k \right\rangle = \int \mathrm{d}^2 p \left( -\frac{\partial f_{\mathrm{eq}}}{\partial \epsilon} \right) \frac{\partial \epsilon}{\partial p_i} p_j p_k = \int \mathrm{d}^2 p \left( \delta_{ij} p_k + \delta_{ik} p_j \right) f_{\mathrm{eq}} = 0, \tag{5.101}$$

where the latter integrals arise due to the observation that the $D_6$ invariant equilibrium distribution can only support nonzero expectation values of functions in the $U_0^+$ irrep. That this integral has to vanish on group theoretic grounds is nontrivial, since a conserved entropy current can be constructed (in the absence of coupling to background gauge fields or geometry) [34].

## 5.3 First-order hydrodynamics

Now, we turn to first-order hydrodynamics. We denote the "incoherent" parts of the currents

$$\left| \tau_{ij}^{\mathrm{inc}} \right\rangle \equiv \left| \tau_{ij} \right\rangle - \frac{\left\langle \rho \middle| \tau_{ij} \right\rangle}{\left\langle \rho \middle| \rho \right\rangle} \left| \rho \right\rangle, \tag{5.102a}$$

$$\left| j_i^{\mathrm{inc}} \right\rangle \equiv \left| j_i \right\rangle - \frac{\left\langle p_i \middle| j_i \right\rangle}{\left\langle p_i \middle| p_i \right\rangle} \left| p_i \right\rangle \tag{5.102b}$$

to be the components of the microscopic currents that do not overlap with hydrodynamic modes. Observe that, e.g.,

$$\mathsf{L}_{ba} \left| \rho \right\rangle = \mathrm{i} k_i \left| j_i^{\mathrm{inc}} \right\rangle. \tag{5.103}$$

By the definition of $\mathsf{W}'$ (5.91), we can therefore immediately deduce the first-order corrections to the equations of motion by evaluating matrix elements of the inverse collision integral $\mathsf{W}^{-1}$ (which is well-defined on nonconserved modes). To do this, it is convenient to first split the stress tensor into its three irreps via

$$\left| \tau_+ \right\rangle \equiv \left| \tau_{xx} \right\rangle + \left| \tau_{yy} \right\rangle, \tag{5.104a}$$

$$\left| \tau_- \right\rangle \equiv \left| \tau_{xy} \right\rangle - \left| \tau_{yx} \right\rangle, \tag{5.104b}$$

$$\left\{ \left| \tau_x \right\rangle, \left| \tau_y \right\rangle \right\} \equiv \left\{ \left| \tau_{xy} \right\rangle + \left| \tau_{yx} \right\rangle, \left| \tau_{xx} \right\rangle - \left| \tau_{yy} \right\rangle \right\}, \tag{5.104c}$$

where, in the last line, we defined $\tau_i = \lambda_{ijk} \tau_{jk}$. Using $\sigma^{\mathrm{inc}} = \chi D$ to denote the incoherent conductivity, we find

$$4\eta_\circ = \langle \tau_- | \mathsf{W}^{-1} | \tau_- \rangle, \tag{5.105a}$$

$$4\zeta = \langle \tau_+^{\mathrm{inc}} | \mathsf{W}^{-1} | \tau_+^{\mathrm{inc}} \rangle, \tag{5.105b}$$

$$4\eta \delta_{ij} = \langle \tau_i | \mathsf{W}^{-1} | \tau_j \rangle, \tag{5.105c}$$

$$\sigma^{\mathrm{inc}} \delta_{ij} = \langle j_i^{\mathrm{inc}} | \mathsf{W}^{-1} | j_j^{\mathrm{inc}} \rangle, \tag{5.105d}$$

$$2\alpha \delta_{ij} = \langle \tau_i | \mathsf{W}^{-1} | j_j^{\mathrm{inc}} \rangle. \tag{5.105e}$$

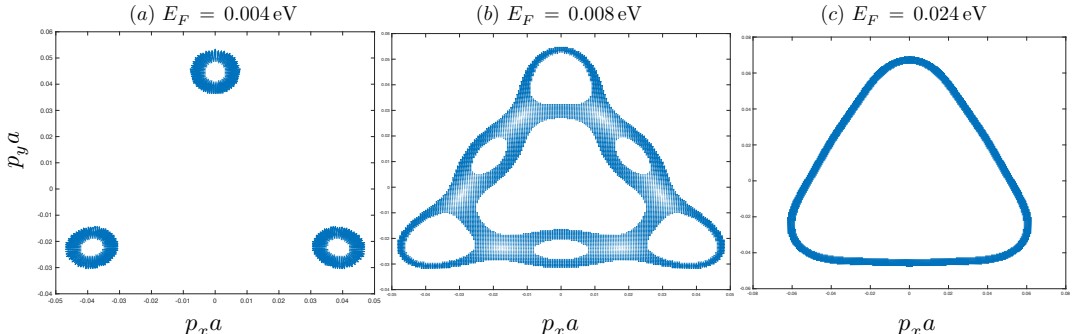

Figure 7: Plots of the Fermi surface in momentum space for ABA trilayer graphene; the three panels correspond to different values of the Fermi energy $E_F$, leading to different Fermi surface geometries and hydrodynamic coefficients. The three values of $E_F$ are 0.004 eV (a), 0.008 eV (b), and 0.024 eV (c); the corresponding hydrodynamic coefficients are reported in Table 2.

Indeed, upon plugging (5.105) into (5.90), we recover (3.28). This demonstrates that (5.105) gives the dissipative coefficients within a kinetic theory. Observe that Onsager reciprocity—which relates $\alpha$ to $\beta$ via (3.34)—is guaranteed by the symmetry of the stress tensor; we will not refer to $\beta$ further in this section. Furthermore, the required positivity conditions on dissipative coefficients are assured by the positive semidefiniteness of W.

Note that we have made the somewhat crude approximation that all eigenvalues of W are either 0 or $\tau_{ee}^{-1}$. Correspondingly, *all* of the dissipative coefficients in (5.105) depend on temperature $T$ through $\tau_{ee}$, e.g., $\eta_\circ \sim W^{-1} \sim \tau_{ee} \sim T^2$ up to subleading $\log T$ corrections. However, in reality, whenever the Fermi surface is a convex polygon, this assumption is likely to break down. The problem is most studied on circular (isotropic) Fermi surfaces [49–52], where one finds that even harmonics of the distribution function are much shorter lived (except for the charge/energy density) while odd harmonics of the distribution function are long lived. However, this notion of even versus odd harmonics stops making sense upon restriction from O(2) to $D_6$: As noted in Sec. 2.1.2, the $k$th harmonic $\mathcal{R}_k$ of O(2) realizes a combination of the scalar irreps $U_0^\pm$ of $D_6$ if $k$ is a multiple of three, and realizes $R_1$ (the vector irrep of $D_6$) otherwise. In particular, note that the odd harmonic $k = 1$ and even harmonic $k = 2$ both map to $R_1$ under $D_6$; more generally, there is no hierachy of even versus odd "harmonics" under $D_6$.

Additionally, in our consideration of trilayer graphene in Sec. 5.4, we note that many of the resulting $D_6$-invariant Fermi surfaces are not convex, so there may be no hierarchy of time scales between distinct types of Fermi surface excitations (see Fig. 7). A more detailed scrutiny of these points would be an interesting direction for future work.

## 5.4 Estimate of $\alpha$ on (half of) ABA trilayer graphene's Fermi surface

It is instructive to determine the relative magnitudes of the dissipative hydrodynamic coefficients in an actual concrete model of a Fermi liquid with a triangular Fermi surface. Note that we consider graphene so as to recover particular values for $\alpha$ relative the other dissipative coefficients in a familiar 2D material; our goal is *not* to consider the rich electron hydrodynamics of graphene (including effects resulting from broken Galilean invariance) [15, 53].

Following [54], we consider the triangular Fermi surfaces that arise near the $K$-point in the Brillouin zone of ABA-trilayer graphene [55]. In ABA trilayer graphene, the inversion-symmetric partner of this Fermi surface will arise elsewhere in the Brillouin zone. Whether or not it is possible to gap out exactly one of these Fermi surfaces to obtain a genuinely triangular Fermi surface, the microscopic band structure for this material (which is numerically computed

Table 2: Dimensionless hydrodynamic coefficients for ABA graphene; columns correspond respectively to the Fermi surfaces plotted in Fig. 7. The last row captures how close a given Fermi surface is to saturating the Onsager constraint (3.36).

| Coeff. \ $E_F$ | 0.004 eV | 0.008 eV | 0.024 eV |
|---|---|---|---|
| $\tilde{\eta}$ | 0.27(5) | 0.35(2) | 0.252(7) |
| $\tilde{\eta}_\circ$ | 0.30(2) | 0.29(0) | 0.0887 |
| $\tilde{\zeta}$ | 0.24(4) | 0.386 | 0.0094 |
| $\tilde{\sigma}^{\text{inc}}$ | 0.50(0) | 0.40(6) | 0.0929 |
| $\tilde{\alpha}$ | 0.056(9) | 0.1394 | 0.138(2) |
| $\tilde{\alpha}^2/\tilde{\eta}\,\tilde{\sigma}^{\text{inc}}$ | 0.023(5) | 0.13(6) | 0.814(0) |

from a six-band model identically to what is described in [54]) represents a useful example for what one *might* expect in other (quasi-) two-dimensional electron liquids.

As in [54], we numerically determine the Fermi surfaces for varying Fermi energies $E_F$; like in graphene, $E_F = 0$ would correspond to a charge neutral point with vanishing Fermi surface, so when $E_F$ is small we can identify small triangular Fermi surfaces. We measure the average Fermi velocity (squared) near the Fermi surface, which we denote as $\langle v_F^2 \rangle$. We then consider kinetic theory in the relaxation time approximation where all nonvanishing eigenvalues of W are given by $1/\tau_{\text{ee}}$. Following (5.105), we then define the *dimensionless* transport coefficients

$$4\tilde{\eta}_\circ = \frac{\langle \tau_- | W^{-1} | \tau_- \rangle}{\langle p_x | p_x \rangle \langle v_F^2 \rangle \tau_{\text{ee}}} \,, \tag{5.106a}$$

$$4\tilde{\zeta} = \frac{\langle \tau_+^{\text{inc}} | W^{-1} | \tau_+^{\text{inc}} \rangle}{\langle p_x | p_x \rangle \langle v_F^2 \rangle \tau_{\text{ee}}} \,, \tag{5.106b}$$

$$4\tilde{\eta}\delta_{ij} = \frac{\langle \tau_i | W^{-1} | \tau_j \rangle}{\langle p_x | p_x \rangle \langle v_F^2 \rangle \tau_{\text{ee}}} \,, \tag{5.106c}$$

$$\tilde{\sigma}^{\text{inc}}\delta_{ij} = \frac{\langle j_i^{\text{inc}} | W^{-1} | j_j^{\text{inc}} \rangle}{\langle \rho | \rho \rangle \langle v_F^2 \rangle \tau_{\text{ee}}} \,, \tag{5.106d}$$

$$2\tilde{\alpha}\delta_{ij} = \frac{\langle \tau_i | W^{-1} | j_j^{\text{inc}} \rangle}{\sqrt{\langle p_x | p_x \rangle \langle \rho | \rho \rangle} \langle v_F^2 \rangle \tau_{\text{ee}}} \,. \tag{5.106e}$$

The right hand side of these equations effectively normalizes each quantity by the typical scale that one expects for these hydrodynamic coefficients: Dividing by $\tau_{\text{ee}}$ effectively removes $W^{-1}$, and the resulting denominator in each fraction corresponds heuristically to the expected susceptibilities of the stress tensor or current operators (we did not use the exact susceptibilities as that would trivially make certain constants equal to one). In each case, we find the constants on the LHS obey, e.g., $0 \le \tilde{\eta} \le 1$. When $\tilde{\eta}_\circ \ll 1$, it signifies that the Fermi surface is too close to circular [16,54]; when $\tilde{\zeta} \ll 1$, it signifies that the Fermi surface is too simple and there are not $U_0^+$ modes that can distort the Fermi surface without modifying the density; when $\tilde{\sigma}^{\text{inc}} \ll 1$, it again signifies the Fermi surface is too close to circular (when it will obtain approximate Galilean invariance [53]). And most importantly, when $\tilde{\alpha} \ll 1$, it signifies that while $D_6$ symmetry is *exact*, the inversion-breaking is (in some sense) weak, and it will be challenging to see the new hydrodynamic coefficients of the $D_6$-invariant fluid.

We now report the results of our calculations and give a sense of the relative scale of $\alpha$

in a material with a $D_6$-invariant Fermi surface, compared to the viscosities and incoherent conductivity. The corresponding Fermi surfaces are plotted in Fig. 7, and the hydrodynamic coefficients are reported in Table 2. We observe that $\alpha$ is generally somewhat smaller than other coefficients, but it is not smaller by multiple orders of magnitude. Thus, it is possible that the effect may be visible in a realistic electron liquid with $D_6$ symmetry.

## 6 Outlook

We have presented the hydrodynamics of a two-dimensional fluid with $D_6$ point group. In contrast to earlier works which focus on the viscosity tensor [16–21], we have additionally found a new dissipative coefficient $\alpha$ (and its Onsager partner $\beta$), which is allowed by the explicit breaking of spatial-inversion and time-reversal symmetries.

We expect that the simplest way to look for $\alpha \neq 0$ in experiments on electron fluids is to use nitrogen-vacancy center magnetometry [8,9] to detect currents at the center of symmetry-exploiting device geometries proposed in Sec. 4. However, it may also be possible to look for this effect in classical soft or active matter made up of microscopic constituents that prefer to be arranged in a "triangular lattice". We are not aware of any liquid crystal molecules that are appropriate for this purpose, but it may be possible to engineer active fluids comprised of triangular objects following [56], which studied parity-breaking active matter fluids made out of spinning disks. We hope to return to this problem in future work.

## Acknowledgements

We thank P. Glorioso, X. Huang, and M. Qi for useful discussions and collaboration on related work. AJF was supported in part by a Simons Investigator Award via Leo Radzihovsky. AL was supported in part by the Alfred P. Sloan Foundation through Grant FG-2020-13795 and through the Gordon and Betty Moore Foundation's EPiQS Initiative via Grant GBMF10279.

## A Stream function and steady state equations

### A.1 Steady-state equations of motion

Starting from the steady state continuity equations, (3.50), we can obtain expressions for the density $\rho$ and streaming function $\psi$ (3.49). The linearized continuity equations take the general form

$$0 = \partial_t \rho + \partial_i j_i, \tag{A.1a}$$

$$0 = \rho_0 \partial_t v_i + \partial_j \tau_{ji}, \tag{A.1b}$$

where

$$j_i \equiv \rho_0 v_i - D \partial_i \rho - \alpha \lambda_{ijk} \partial_j v_k, \tag{A.2a}$$

$$\tau_{ij} = p \delta_{ij} - \eta_{ijkl} \partial_k v_l - \beta \lambda_{ijk} \partial_k \rho, \tag{A.2b}$$

$$\eta_{ijkl} = \eta \left( \sigma_{ij}^x \sigma_{kl}^x + \sigma_{ij}^z \sigma_{kl}^z \right) + \zeta \delta_{ij} \delta_{kl} + \eta_\circ \epsilon_{ij} \epsilon_{kl}, \tag{A.2c}$$

$$\lambda_{ijk} = \delta_{k,1} \sigma_{ij}^x + \delta_{k,2} \sigma_{ij}^z, \tag{A.2d}$$

and $p = c^2 \rho$. We then write the continuity equations for charge and momentum as

$$\partial_t \rho = -\nabla \cdot j = -\rho_0 \left( \nabla \cdot v \right) + D \nabla^2 \rho + 2\alpha \partial_x \partial_y v_x + \alpha \left( \partial_x^2 - \partial_y^2 \right) v_y \,, \tag{A.3a}$$

$$\rho_0 \partial_t v_i = -c^2 \partial_i \rho + \eta \partial_j \partial_j v_i + \zeta \partial_i \partial_j v_j + \eta_\circ \partial_j \epsilon_{ji} \partial_k \epsilon_{kl} v_l + \beta \lambda_{ijk} \partial_j \partial_k \rho \,, \tag{A.3b}$$

and we can also write $\partial_j \epsilon_{ji} = -\epsilon_{ij} \partial_j$ for convenience.

We now take the steady state limit ($\partial_t \rho = \partial_t v_i = 0$), which implies that $\nabla \cdot j = 0$, and in any continuous two-dimensional space, this implies that the current is the curl of some scalar field,

$$j_i = \rho_0 \epsilon_{ij} \partial_j \psi \,, \tag{A.4}$$

where $\psi$ is the "stream function" and $j_x = \rho_0 \partial_y \psi$ and $j_y = -\rho_0 \partial_x \psi$. We can relate the velocity to the stream function via

$$v_x = \partial_y \psi + \frac{D}{\rho_0} \partial_x \rho + \frac{\alpha}{\rho_0} \left( \partial_x v_y + \partial_y v_x \right) \,, \tag{A.5a}$$

$$v_y = -\partial_x \psi + \frac{D}{\rho_0} \partial_y \rho + \frac{\alpha}{\rho_0} \left( \partial_x v_x - \partial_y v_y \right) \,, \tag{A.5b}$$

and we also have

$$\nabla \cdot v = \frac{D}{\rho_0} \nabla^2 \rho + 2 \frac{\alpha}{\rho_0} \partial_x \partial_y v_x + \frac{\alpha}{\rho_0} \left( \partial_x^2 - \partial_y^2 \right) v_y \,, \tag{A.6a}$$

$$\nabla \times v = -\nabla^2 \psi + \frac{\alpha}{\rho_0} \left( \partial_x^2 - \partial_y^2 \right) v_x - 2 \frac{\alpha}{\rho_0} \partial_x \partial_y v_y \,, \tag{A.6b}$$

and we can recursively insert the expressions for $v_{x,y}$ to recover equations for $v_j$ (along with $\nabla \cdot v$ and $\nabla \times v$) in terms of $\rho$ and $\psi$ alone, to arbitrary order in $\ell_{\mathrm{ee}}$.

We find equations of motion for the density $\rho$ and stream function $\psi$ by taking the divergence and curl of (A.3b), given respectively by

$$c^2 \nabla^2 \rho = (\eta + \zeta) \nabla^2 \left( \nabla \cdot v \right) + \beta \lambda_{ijk} \partial_i \partial_j \partial_k \rho \,, \tag{A.7a}$$

$$0 = \left( \eta + \eta_\circ \right) \nabla^2 \left( \nabla \times v \right) + \beta \partial_i \epsilon_{ij} \lambda_{jkl} \partial_k \partial_l \rho \,, \tag{A.7b}$$

which we can simplify using the expressions for the divergence (A.6a) and curl (A.6b) of the velocity. Using these results, we now write the original continuity equations in terms of $\rho$ and $\psi$ alone.

Starting from the modified biharmonic equation,

$$\nabla^4 \psi = 2 \xi \nabla^2 \left( 3\partial_x^2 - \partial_y^2 \right) \partial_y \psi \,, \tag{3.52}$$

we seek perturbative solutions in $\xi$. To lowest order, the solutions are of the form

$$\psi(x, y) = \psi_0(x, y) + \xi \psi_1(x, y) \,, \tag{3.53}$$

where the unperturbed piece $\psi_0$ satisfies the usual biharmonic equation,

$$\nabla^4 \psi_0 = 0 \,, \tag{A.8}$$

and the perturbation $\psi_1$ satisfies

$$\nabla^4 \psi_1 = 2 \nabla^2 \left( 3\partial_x^2 - \partial_y^2 \right) \partial_y \psi_0 \,. \tag{A.9}$$

We now recover the particular solutions $\psi_{0,1}$ relevant to the hexagonal device experiment proposed in Sec. 4.1.

## A.2 Equation for the density

Taking the divergence of (A.3b) recovers

$$0 = -c^2\nabla^2\rho + (\eta+\zeta)\nabla^2(\nabla\cdot v) + \beta\lambda_{ijk}\partial_i\partial_j\partial_k\rho\,,$$

$$c^2\nabla^2\rho = (\eta+\zeta)\nabla^2\nabla\cdot v + \beta\left(3\partial_x^2 - \partial_y^2\right)\partial_y\rho\,, \tag{A.10}$$

$$c^2\nabla^2\rho = (\eta+\zeta)\nabla^2\left(\frac{D}{\rho_0}\nabla^2\rho + \frac{\alpha}{\rho_0}\left(3\partial_y^2 - \partial_x^2\right)\partial_x\psi\right) + \beta\left(3\partial_x^2 - \partial_y^2\right)\partial_y\rho\,,$$

and ignoring the $O(\ell_{ee}^2)$ terms— containing two factors of the $O(\ell_{ee})$ coefficients $\eta, \zeta, D, \alpha$—we find

$$\nabla^2\rho = \frac{\beta}{c^2}\left(3\partial_x^2 - \partial_y^2\right)\partial_y\rho = \xi\left(3\partial_x^2 - \partial_y^2\right)\partial_y\rho\,, \tag{A.11}$$

and in the limit $\alpha = \beta = 0$, corresponding to hexagonal rotation symmetry (point group $D_{12}$), we recover

$$\nabla^2\rho = 0\,, \tag{A.12}$$

in agreement with Ref. [17].

## A.3 Equation for the stream function

Taking the curl of (A.3b) (i.e., multiplying the $j$th component by $\partial_i\epsilon_{ij}$ from the left) recovers

$$0 = -c^2\partial_i\epsilon_{ij}\partial_j\rho + \eta\nabla^2\partial_i\epsilon_{ij}v_j + \zeta\partial_i\epsilon_{ij}\partial_j\partial_k v_k - \partial_i\epsilon_{ij}\eta_\circ\epsilon_{jk}\partial_k\partial_m\epsilon_{mn}v_n + \beta\partial_i\epsilon_{ij}\lambda_{jkl}\partial_k\partial_l\rho\,, \tag{A.13}$$

and we note that $\partial_i\epsilon_{ij}\partial_j = 0$ and $\epsilon_{ij}\epsilon_{jk} = -\delta_{ik}$, and the above becomes

$$\begin{aligned}
0 &= (\eta+\eta_\circ)\nabla^2\partial_i\epsilon_{ij}v_j + \beta\partial_i\epsilon_{ij}\left(\delta_{j,1}\sigma_{kl}^x + \delta_{j,2}\sigma_{kl}^z\right)\partial_k\partial_l\rho \\
&= (\eta+\eta_\circ)\nabla^2\left(\partial_i\epsilon_{ij}v_j\right) + \beta\left(-\partial_y\sigma_{kl}^x + \partial_x\sigma_{kl}^z\right)\partial_k\partial_l\rho \\
&= (\eta+\eta_\circ)\nabla^2(\nabla\times v) + \beta\left(\partial_x^2 - 3\partial_y^2\right)\partial_x\rho\,,
\end{aligned} \tag{A.14}$$

and we simplify this expression using the following relations:

$$\begin{aligned}
\nabla\times v &= -\nabla^2\psi + \frac{\alpha}{\rho_0}\left(\partial_x^2 - \partial_y^2\right)v_x - 2\frac{\alpha}{\rho_0}\partial_x\partial_y v_y \\
&= -\nabla^2\psi + \frac{\alpha}{\rho_0}\left(\partial_x^2 - \partial_y^2\right)\left[\partial_y\psi + \dots\right] - 2\frac{\alpha}{\rho_0}\partial_x\partial_y\left[-\partial_x\psi + \dots\right] \\
&= -\nabla^2\psi + \frac{\alpha}{\rho_0}\left(3\partial_x^2 - \partial_y^2\right)\psi + O\left(\ell_{ee}^2\right)\,,
\end{aligned} \tag{A.15}$$

$$\begin{aligned}
c^2\partial_x\rho &= \eta\nabla^2 v_x + \zeta\partial_x(\nabla\cdot v) - \eta_\circ\partial_y(\nabla\times v) + 2\beta\partial_x\partial_y\rho \\
&= \eta\nabla^2\left[\partial_y\psi + \dots\right] + \zeta\partial_x O\left(\ell_{ee}\right) - \eta_\circ\partial_y\left[-\nabla^2\psi + \dots\right] + 2\beta\partial_x\partial_y\rho\,, \\
c^2\partial_x\rho &= (\eta+\eta_\circ)\nabla^2\partial_y\psi + O\left(\ell_{ee}^2\right)\,,
\end{aligned} \tag{A.16}$$

where we note that $\partial_x\rho$ is $O(\ell_{ee})$, so if we insert the expression for $\partial_x\rho$ into the $\beta\partial_x\rho$ term, all new terms will be $O(\ell_{ee}^2)$, and therefore subleading. We find similarly for $\partial_y\rho$,

$$\begin{aligned}
c^2\partial_y\rho &= \eta\nabla^2 v_y + \zeta\partial_y(\nabla\cdot v) + \eta_\circ\partial_x(\nabla\times v) + \beta\left(\partial_x^2 - \partial_y^2\right)\rho \\
&= \eta\nabla^2\left[-\partial_x\psi + \dots\right] + \zeta\partial_y O\left(\ell_{ee}\right) + \eta_\circ\partial_x\left[-\nabla^2\psi + \dots\right] + \beta\left(\partial_x^2 - \partial_y^2\right)\rho\,, \\
c^2\partial_y\rho &= -(\eta+\eta_\circ)\nabla^2\partial_x\psi + O\left(\ell_{ee}^2\right)\,,
\end{aligned} \tag{A.17}$$

and using this, the final equation for the stream function $\psi$ is

$$\nabla^4 \psi = \frac{\alpha}{\rho_0} \nabla^2 \left( 3\partial_x^2 - \partial_y^2 \right) \partial_y \psi + \frac{\beta}{c^2} \nabla^2 \left\{ 2\partial_x^2 \partial_y + \left( \partial_x^2 - \partial_y^2 \right) \partial_y \right\} \psi, \tag{A.18}$$

and the final result is

$$\nabla^4 \psi = \left( \frac{\alpha}{\rho_0} + \frac{\beta}{c^2} \right) \nabla^2 \left( 3\partial_x^2 - \partial_y^2 \right) \partial_y \psi = 2\,\xi\, \nabla^2 \left( 3\partial_x^2 - \partial_y^2 \right) \partial_y \psi. \tag{3.52}$$

## A.4  Complex coordinates

We note that (3.52) is easier to solve in complex coordinates,

$$z = x + i\,y\,,\ \ \bar{z} = x - i\,y\,, \quad \text{with} \quad x = \frac{1}{2}(z + \bar{z})\,,\ \ y = \frac{1}{2i}(z - \bar{z})\,, \tag{A.19}$$

where the various derivatives are related by

$$\partial_x = \partial_z + \partial_{\bar{z}}\,,\ \ \partial_y = i\left( \partial_z - \partial_{\bar{z}} \right)\,, \quad \text{and} \quad \partial_z = \frac{1}{2}\left( \partial_x - i\partial_y \right)\,,\ \ \partial_{\bar{z}} = \frac{1}{2}\left( \partial_x + i\partial_y \right)\,, \tag{A.20}$$

and the relevant derivative operators can be neatly written as:

$$\nabla^2 = 4\,\partial_z\,\partial_{\bar{z}}\,, \tag{A.21a}$$

$$\left( 3\,\partial_x^2 - \partial_y^2 \right) \partial_y = 4i \left( \partial_z^3 - \partial_{\bar{z}}^3 \right)\,, \tag{A.21b}$$

and the modified biharmonic equation takes the form

$$\partial_z \partial_{\bar{z}} \left\{ \partial_z \partial_{\bar{z}} - 2\,i\,\xi \left( \partial_z^3 - \partial_{\bar{z}}^3 \right) \right\} \psi(z, \bar{z}) = 0\,, \tag{A.22}$$

where we have dropped an overall factor of 16 for obvious reasons. Note that the combination of $\partial_z^3$ and $\partial_{\bar{z}}^3$ (with prefactor $\xi$), acts on the $m$th harmonic by shifting it to $m - 3$.

## A.5  Polar coordinates

It is also useful to consider (3.51) and (3.52)—as well as their solutions—in polar coordinates,

$$x = r\,\cos\theta\,,\ \ y = r\,\sin\theta\,,\ \ \partial_x = \cos\theta\,\partial_r - \frac{1}{r}\sin\theta\,\partial_\theta\,,\ \ \partial_y = \sin\theta\,\partial_r + \frac{1}{r}\cos\theta\,\partial_\theta\,, \tag{A.23}$$

from which we also have

$$\partial_x^2 = \cos^2\theta\,\partial_r^2 - \frac{1}{r}\sin 2\theta\,\partial_r\partial_\theta + \frac{1}{r^2}\sin^2\theta\,\partial_\theta^2 + \frac{1}{r}\sin^2\theta\,\partial_r + \frac{1}{r^2}\sin 2\theta\,\partial_\theta\,, \tag{A.24a}$$

$$\partial_y^2 = \sin^2\theta\,\partial_r^2 + \frac{1}{r}\sin 2\theta\,\partial_r\partial_\theta + \frac{1}{r^2}\cos^2\theta\,\partial_\theta^2 + \frac{1}{r}\cos^2\theta\,\partial_r - \frac{1}{r^2}\sin 2\theta\,\partial_\theta\,, \tag{A.24b}$$

from which we surmise

$$\nabla^2 = \partial_i\partial_i = \partial_x^2 + \partial_y^2 = \partial_r^2 + \frac{1}{r^2}\partial_\theta^2 + \frac{1}{r}\partial_r\,, \tag{A.25}$$

and also

$$3\partial_x^2 - \partial_y^2 = \frac{\sin 3\theta}{\sin\theta}\,\partial_r^2 - \frac{4}{r}\sin 2\theta\,\partial_r\partial_\theta - \frac{1}{r}\frac{\cos 3\theta}{\cos\theta}\,\partial_r + \frac{4}{r^2}\sin 2\theta\,\partial_\theta\,, \tag{A.26}$$

where we used

$$\cos 3\theta = \left( \cos^2\theta - 3\sin^2\theta \right)\cos\theta\,, \quad \text{and} \quad \sin 3\theta = \left( 3\cos^2\theta - \sin^2\theta \right)\sin\theta\,. \tag{A.27}$$

Taking advantage of the fact that partial derivatives commute, we apply $\partial_y$ to (A.26) to get

$$\left(3\partial_x^2 - \partial_y^2\right)\partial_y = \sin 3\theta \left\{\partial_r^3 - \frac{3}{r^2}\partial_r\partial_\theta^2 - \frac{3}{r}\partial_r^2 + \frac{6}{r^3}\partial_\theta^2 + \frac{3}{r^2}\partial_r\right\}$$
$$+ \cos 3\theta \left\{\frac{3}{r}\partial_r^2\partial_\theta - \frac{1}{r^3}\partial_\theta^3 - \frac{9}{r^2}\partial_r\partial_\theta + \frac{8}{r^3}\partial_\theta\right\}, \tag{A.28}$$

which is invariant under $\theta \to \theta + 2\pi/3$, consistent with the $\text{D}_6$ rotational symmetry of the Fermi surface.

The biharmonic operator $\nabla^4 = (\nabla^2)^2$ takes the polar form

$$\nabla^4 = \partial_r^4 + \frac{2}{r^2}\partial_r^2\partial_\theta^2 + \frac{1}{r^4}\partial_\theta^4 + \frac{2}{r}\partial_r^3 - \frac{2}{r^3}\partial_r\partial_\theta^2 + \frac{4}{r^4}\partial_\theta^2 - \frac{1}{r^2}\partial_r^2 + \frac{1}{r^3}\partial_r, \tag{A.29}$$

in agreement with [46].

Finally, in polar coordinates the current has components

$$j_r = \hat{\boldsymbol{r}} \cdot j = \cos(\theta)j_x + \sin(\theta)j_y = \rho_0\left(\cos(\theta)\partial_y - \sin(\theta)\partial_x\right)\psi = \frac{1}{r}\rho_0\,\partial_\theta\,\psi, \tag{A.30a}$$

$$j_\theta = \hat{\boldsymbol{\theta}} \cdot j = -\sin(\theta)j_x + \cos(\theta)j_y = -\rho_0\left(\sin(\theta)\partial_y + \cos(\theta)\partial_x\right)\psi = -\rho_0\,\partial_r\,\psi. \tag{A.30b}$$

## A.6 General solution to the biharmonic equation

Ignoring the perturbing terms on the RHS of (3.52), the general solution to the biharmonic equation in polar coordinates is well known [46], given by

$$\psi(r,\theta) = a_0\,r^2 + b_0\,r^2\ln r + c_0 + d_0\ln r + \left(a_0'\,r^2 + b_0'\,r^2\ln r + c_0' + d_0'\ln r\right)\theta$$
$$+ \left(a_1\,r + b_1\,r^3 + c_1\,r^{-1} + d_1\,r\ln r\right)\cos\theta + \left(a_1'\,r + b_1'\,r^3 + c_1'\,r^{-1} + d_1'\,r\ln r\right)\sin\theta$$
$$+ \left(A_1\,r + D_1\,r\ln r\right)\theta\cos\theta + \left(A_1'\,r + D_1'\,r\ln r\right)\theta\sin\theta$$
$$+ \sum_{m=2}^{\infty}\left(a_m\,r^m + b_m\,r^{m+2} + c_m\,r^{-m} + d_m\,r^{2-m}\right)\cos(m\theta)$$
$$+ \sum_{m=2}^{\infty}\left(a_m'\,r^m + b_m'\,r^{m+2} + c_m'\,r^{-m} + d_m'\,r^{2-m}\right)\sin(m\theta). \tag{A.31}$$

For the components of the current $j_i = \rho_0\epsilon_{ij}\partial_j\psi$ to be nonsingular at the center of the device ($r = 0$), many of the coefficients in (A.31) must be zero. Additionally, the terms with a factor of $\theta$ correspond to undesirable rotating solutions that are discontinuous as $\theta$ passes through $2\pi$ (i.e., the discontinuity emerges because these are not periodic functions of $\theta$), and we set these coefficients to zero as well.

The remaining, physical terms are given in polar coordinates by

$$\psi_0(r,\theta) = a_0\,r^2 + b_0\,r^2\ln r + \sum_{m=1}^{\infty}\left\{\left(a_m + b_m\,r^2\right)r^m\cos(m\theta) + \left(a_m' + b_m'\,r^2\right)r^m\sin(m\theta)\right\}, \tag{3.55}$$

or in terms of complex coordinates as

$$\psi_0(z,\bar{z}) = a_0\,\bar{z}z + \frac{1}{2}\,b_0\,\bar{z}z\ln(\bar{z}z) + \sum_{m=1}^{\infty}\left\{\left(a_m + b_m\,\bar{z}z\right)\frac{(z^m + \bar{z}^m)}{2} + \left(a_m' + b_m'\,\bar{z}z\right)\frac{(z^m - \bar{z}^m)}{2i}\right\}, \tag{A.32}$$

where we have absorbed the $m = 1$ harmonic into the infinite sum.

### A.7 Solution to leading order in $\xi$

Equipped with a general solution to the unperturbed problem (3.55), we will now determine $\psi_1$ by solving

$$\nabla^4\psi_1 = 2\nabla^2\left(3\partial_x^2 - \partial_y^2\right)\partial_y\psi_0,\tag{3.52}$$

which is most conveniently accomplished in complex coordinates, where the above takes the tidy form

$$16\,\partial_z^2\,\partial_{\bar z}^2\,\psi_1(z,\bar z) = 32\,i\left(\partial_z^3 - \partial_{\bar z}^3\right)\partial_z\,\partial_{\bar z}\,\psi_0,\tag{A.33}$$

with $\psi_0$ given by (A.32), so that (A.33) becomes,

$$\partial_z^2\,\partial_{\bar z}^2\,\psi_1 = \frac{4\,b_0}{(\bar z z)^3}\frac{z^3-\bar z^3}{2i} + 2\sum_{m=3}^{\infty}\frac{(m+1)!}{(m-3)!}\left\{b'_m\frac{z^{m-3}+\bar z^{m-3}}{2} - b_m\frac{z^{m-3}-\bar z^{m-3}}{2i}\right\},\tag{A.34}$$

and particular solutions—corresponding to each term on the RHS of (A.34) above—can be found using Mathematica. The resulting particular is

$$\psi_1(z,\bar z) = \frac{b_0}{z\bar z}\frac{z^3-\bar z^3}{2i} + 12\,b'_3\,z^2\bar z^2 + z^2\bar z^2\sum_{m=4}^{\infty}m(m+1)\left\{b'_m\frac{z^{m-3}+\bar z^{m-3}}{2} - b_m\frac{z^{m-3}-\bar z^{m-3}}{2i}\right\},\tag{A.35}$$

and in polar coordinates,

$$\psi_1(r,\theta) = b_0 r\,\sin(3\theta) + r^4\sum_{n=0}^{\infty}(n+3)(n+4)\,r^n\left\{b'_{n+3}\cos[n\theta] - b_{n+3}\sin[n\theta]\right\},\tag{A.36}$$

and we can add to this any biharmonic functions (i.e., any of the terms appearing in (3.55)), giving for the general solution at first order

$$\psi_1(r,\theta) = b_0 r\,\sin(3\theta) + r^4\sum_{n=0}^{\infty}(n+3)(n+4)\,r^n\left\{b'_{n+3}\cos[n\theta] - b_{n+3}\sin[n\theta]\right\}\tag{A.37}$$

$$+\tilde a_0\,r^2 + \tilde b_0\,r^2\ln r + \sum_{m=1}^{\infty}\left\{\left(\tilde a_m + \tilde b_m\,r^2\right)r^m\cos(m\theta) + \left(\tilde a'_m + \tilde b'_m\,r^2\right)r^m\sin(m\theta)\right\},$$

where the coefficients with twiddles should not be confused with the same set of terms that appear in the zeroth order solution.

### A.8 Current in polar coordinates

In polar coordinates, the radial and angular components of the current are given by

$$j_r = \frac{1}{r}\rho_0\,\partial_\theta\,\psi,\quad\text{and}\quad j_\theta = -\rho_0\,\partial_r\,\psi,\tag{A.38}$$

and writing $j(r,\theta) = j^{(0)}(r,\theta) + \xi\,j^{(1)}(r,\theta)$, the zeroth order current is given by

$$j_r^{(0)}(r,\theta) = \frac{\rho_0}{r}\partial_\theta\psi_0$$

$$= \rho_0\sum_{m=1}^{\infty}m\,r^{m-1}\left\{\left(a'_m + r^2\,b'_m\right)\cos(m\theta) - \left(a_m + r^2\,b_m\right)\sin(m\theta)\right\},\tag{A.39a}$$

$$j_\theta^{(0)}(r,\theta) = -\rho_0\,\partial_r\psi_0$$

$$= -\rho_0 r\left\{2a_0 + (1+2\ln r)b_0\right\} - \rho_0\sum_{m=1}^{\infty}m\,r^{m-1}\left\{a_m\cos(m\theta) + a'_m\sin(m\theta)\right\}$$

$$- \rho_0\sum_{m=1}^{\infty}(m+2)\,r^{m+1}\left\{b_m\cos(m\theta) + b'_m\sin(m\theta)\right\},\tag{A.39b}$$

and the first order correction is given by

$$
\begin{aligned}
j_r^{(1)}(r,\theta) &= \frac{\rho_0}{r}\,\partial_\theta \psi_1 \\
&= 3\rho_0 b_0 \cos(3\theta) - \rho_0 r^3 \sum_{n=1}^{\infty} n\,(n+3)(n+4)\,r^n \left\{ b_{n+3}\cos[n\theta] + b'_{n+3}\sin[n\theta] \right\} \\
&\quad + \rho_0 \sum_{m=1}^{\infty} m\,r^{m-1} \left\{ \left(\tilde{a}'_m + r^2 \tilde{b}'_m\right)\cos(m\theta) - \left(\tilde{a}_m + r^2 \tilde{b}_m\right)\sin(m\theta) \right\}, \qquad \text{(A.40a)}
\end{aligned}
$$

$$
\begin{aligned}
j_\theta^{(1)}(r,\theta) &= -\rho_0\,\partial_r \psi_1 \\
&= -\rho_0 b_0 \sin(3\theta) - 48\rho_0 b'_3 r^3 \\
&\quad + \rho_0 r^3 \sum_{n=1}^{\infty} (n+3)\,(n+4)^2\,r^n \left\{ b_{n+3}\sin[n\theta] - b'_{n+3}\cos[n\theta] \right\} \\
&\quad - \rho_0 r \left\{ 2\tilde{a}_0 + (1+2\ln r)\tilde{b}_0 \right\} - \rho_0 \sum_{m=1}^{\infty} m\,r^{m-1} \left\{ \tilde{a}_m \cos(m\theta) + \tilde{a}'_m \sin(m\theta) \right\} \\
&\quad - \rho_0 \sum_{m=1}^{\infty} (m+2)\,r^{m+1} \left\{ \tilde{b}_m \cos(m\theta) + \tilde{b}'_m \sin(m\theta) \right\}. \qquad \text{(A.40b)}
\end{aligned}
$$

## B  Details for the hexagonal device

Starting from the equation of motion for the stream function, (3.52), we derive solutions to leading order in $\xi = \alpha/\rho_0 = \beta/c^2$, which is $O(\ell_{\text{ee}})$. The stream function relates to the current at the device center according to

$$
j_i = \rho_0 \epsilon_{ij} \partial_j \psi, \qquad (3.49)
$$

where $\psi$ is the stream function.

### B.1  Current from the leads

Looking at (3.58), the current at the origin of a $D_6$ fluid with arbitrary boundary conditions is given by

$$
\begin{aligned}
j_x(r\to 0) &= \rho_0 a_1 + \rho_0 \xi\left(\tilde{a}'_1 + b_0 \pm 2 b_0\right), &\qquad \text{(4.64a)} \\
j_y(r\to 0) &= -\rho_0\left(a_1 + \xi\,\tilde{a}_1\right), &\qquad \text{(4.64b)}
\end{aligned}
$$

where the $\pm$ above depends on whether $x\to 0$ is taken first ($+$) or $y\to 0$ is taken first ($-$). Hence for the current to be well defined everywhere, we must have $b_0 = 0$, and this will be the case for the boundary conditions of interest.

Note that the $m$th harmonic term in $\psi(r,\theta)$ corresponds to the $\mathcal{R}_m$ irrep of $O(2)$. Regarding Fig. 4, we conclude that the $\theta$ dependence of the current at the boundary,

$$
j_\perp(\theta) \equiv j_r(R,\theta), \quad \text{and} \quad j_\parallel(\theta) \equiv j_\theta(R,\theta), \qquad \text{(B.1)}
$$

must correspond to the $R_2$ irrep of $D_{12}$ (or restrictions of irreps of $O(2)$ to $D_{12}$ corresponding to $R_2$). The harmonics with $m$ even but *not* a multiple of three correspond to irreps of $O(2)$ that reduce to the $R_2$ irrep of $O(2)$; therefore, only these harmonics can be nonzero.

The general functional form of $j_\perp(\theta)$ and $j_\parallel(\theta)$, constrained to the allowed harmonics, is given by

$$j_\perp(\theta) = \rho_0 \sum_{\substack{m \geq 2 \\ m \text{ even} \\ m \bmod 3 \neq 0}} m R^{m-1} \left\{ \left( a'_m + R^2 b'_m \right) \cos(m\theta) - \left( a_m + R^2 b_m \right) \sin(m\theta) \right\}, \tag{B.2a}$$

$$j_\parallel(\theta) = -\rho_0 \sum_{\substack{m \geq 2 \\ m \text{ even} \\ m \bmod 3 \neq 0}} R^{m-1} \left\{ \left( m a_m + (m+2) R^2 b_m \right) \cos(m\theta) + \left( m a'_m + (m+2) R^2 b'_m \right) \sin(m\theta) \right\},$$
$$\tag{B.2b}$$

with $a_m, b_m, a'_m, b'_m$ all zero unless $m$ is even *and* $m$ not a multiple of three. We note that this precludes $m = 0$ terms, including the undesirable coefficient $b_0$. We match the boundary currents $j_\perp$ and $j_\parallel$ (which encode the current in the leads) to $j_r^{(0)}$ and $j_\theta^{(0)}$ only; thus, $j_r^{(1)}$ and $j_\theta^{(1)}$ vanish at the boundaries.

We consider the arrangement of leads depicted in the right panel of Fig. 4 and assume that the wires are infinitesimally thin. We then model the current through the leads via delta functions,

$$j_\perp(\theta) = j_{\text{in}} \left( \delta\left(\theta - 2\pi/3\right) + \delta\left(\theta - 5\pi/3\right) - \delta\left(\theta - \pi/3\right) - \delta\left(\theta - 4\pi/3\right) \right), \quad j_\parallel(\theta) = 0, \tag{B.3}$$

corresponding to current normal to the surface flowing in at 60 and 240 degrees and flowing out at 120 and 300 degrees (measured counterclockwise from the positive $x$ axis), and no current in the $\hat{\theta}$ direction at the boundaries.

## B.2 Resulting current in the device

Using (B.1) and the functional form of $j_{r,\theta}^{(0)}$ from App. A.8, we match the angular and radial current components to the boundary functions (B.1), and integrate both sides against $\cos(m\theta)$ and $\sin(m\theta)$ from $-\pi$ to $\pi$ to fix the coefficients in $j_{r,\theta}^{(0)}$. The resulting relations are

$$-\rho_0 R \left( 2 a_0 + (1 + 2\ln R) b_0 \right) = 0, \tag{B.4a}$$

$$\rho_0 m R^{m-1} \left( a'_m + R^2 b'_m \right) = 0, \tag{B.4b}$$

$$-\rho_0 m R^{m-1} \left( a_m + R^2 b_m \right) = \frac{2}{\pi} j_{\text{in}} \left( \sin\left(\frac{2\pi m}{3}\right) - \sin\left(\frac{\pi m}{3}\right) \right), \tag{B.4c}$$

$$-\rho_0 R^{m-1} \left( m a_m + (m+2) R^2 b_m \right) = 0, \tag{B.4d}$$

$$\rho_0 R^{m-1} \left( m a'_m + (m+2) R^2 b'_m \right) = 0, \tag{B.4e}$$

from which we conclude that

$$a'_m = b'_m = 0. \tag{B.5}$$

Using the fact that

$$\sin\left(\frac{\pi m}{3}\right) - \sin\left(\frac{2\pi m}{3}\right) = (1 + (-1)^m) \sin\left(\frac{\pi m}{3}\right),$$

the other coefficients are given by

$$a_m = \frac{j_{\text{in}}}{\pi \rho_0} \frac{m+2}{m} \frac{(1 + (-1)^m)}{R^{m-1}} \sin\left(\frac{m\pi}{3}\right), \tag{B.6a}$$

$$b_m = -\frac{j_{\text{in}}}{\pi \rho_0} \frac{(1 + (-1)^m)}{R^{m+1}} \sin\left(\frac{m\pi}{3}\right), \tag{B.6b}$$

where $\sin(m\pi/3)$ is zero if $m$ is a multiple of three, and $(1+(-1)^m)$ is zero unless $m$ is even, as required by representation theory.

The zeroth order currents are given by

$$
\begin{aligned}
j_r^{(0)}(r,\theta) &= -\rho_0 \sum_{m=1}^{\infty} m\, r^{m-1} \left(a_m + r^2 b_m\right) \sin(m\theta) \\
&= -\frac{4j_{\text{in}}}{\pi} \sum_{m=1}^{\infty} \left(\frac{r}{R}\right)^{2m-1} \left(1 + m\left(1 - \frac{r^2}{R^2}\right)\right) \sin\left(\frac{2\pi m}{3}\right) \sin(2m\theta),
\end{aligned} \tag{B.7a}
$$

$$
\begin{aligned}
j_\theta^{(0)}(r,\theta) &= -\rho_0 \sum_{m=1}^{\infty} r^{m-1} \left(m\, a_m + (m+2)\, r^2\right) \cos(m\theta) \\
&= -\frac{4j_{\text{in}}}{\pi} \sum_{m=1}^{\infty} (m+1) \left(\frac{r}{R}\right)^{2m-1} \left(1 - \frac{r^2}{R^2}\right) \sin\left(\frac{2\pi m}{3}\right) \cos(2m\theta),
\end{aligned} \tag{B.7b}
$$

where we took $m \to 2m$ since only even $m$ are permitted. Note that both components of the zeroth order current vanish as $r \to 0$. Moving to first order, we have the particular solutions

$$
\begin{aligned}
j_r^{(1)}(r,\theta) &= -\frac{j_{\text{in}}}{\pi R} \sum_{n=1}^{\infty} (1-(-1)^n)\, n\,(n+3)(n+4) \left(\frac{r}{R}\right)^{n+3} \sin\left(\frac{\pi n}{3}\right) \cos(n\theta) \\
&\quad + \rho_0 \sum_{n=1}^{\infty} n\, r^{n-1} \left(\tilde{a}_n' + r^2 \tilde{b}_n'\right) \cos(n\theta),
\end{aligned} \tag{B.8a}
$$

$$
\begin{aligned}
j_\theta^{(1)}(r,\theta) &= \frac{j_{\text{in}}}{\pi R} \sum_{n=1}^{\infty} (1-(-1)^n)(n+3)(n+4)^2 \left(\frac{r}{R}\right)^{n+3} \sin\left(\frac{\pi n}{3}\right) \sin(n\theta) \\
&\quad - \rho_0 \sum_{m=1}^{\infty} r^{m-1} \left\{m\, \tilde{a}_m' + (m+2)\, r^2 \tilde{b}_m'\right\} \sin(m\theta).
\end{aligned} \tag{B.8b}
$$

Because all boundary conditions were satisfied at zeroth order (since the leads do not know about $\xi \neq 0$), both components of the current must vanish at the boundary. This means that the coefficients $\tilde{a}_m = \tilde{b}_m = 0$ for all $m$, while the coefficients $\tilde{a}_m'$ and $\tilde{b}_m'$ have been included above to cancel out the particular solutions at the boundary.

Matching boundary conditions at first order requires that

$$
n\left(\tilde{a}_n' + R^2 \tilde{b}_n'\right) = \frac{j_{\text{in}}}{\pi \rho_0 R^n} (1-(-1)^n)\, n\,(n+3)(n+4) \sin\left(\frac{\pi n}{3}\right), \tag{B.9a}
$$

$$
n\left(\tilde{a}_n' + R^2 \tilde{b}_n'\right) + 2R^2 \tilde{b}_n' = \frac{j_{\text{in}}}{\pi \rho_0 R^n} (1-(-1)^n)(n+3)(n+4)^2 \sin\left(\frac{\pi n}{3}\right), \tag{B.9b}
$$

and we can solve for these coefficients to find

$$
\tilde{a}_n' = -\frac{j_{\text{in}}}{\pi \rho_0 R^n} (1-(-1)^n)(n+3)(n+4) \sin\left(\frac{\pi n}{3}\right),
$$

$$
\tilde{b}_n' = \frac{2j_{\text{in}}}{\pi \rho_0 R^{n+2}} (1-(-1)^n)(n+3)(n+4) \sin\left(\frac{\pi n}{3}\right), \tag{B.10a}
$$

and so the full current to $O(\xi)$ is given by

$$
j_r(r, \theta) = -\frac{4 j_{\text{in}}}{\pi} \sum_{m=1}^{\infty} \left(\frac{r}{R}\right)^{2m-1} \left(1 + m\left(1 - \frac{r^2}{R^2}\right)\right) \sin\left(\frac{2\pi m}{3}\right) \sin(2m\theta)
$$

$$
- \frac{\xi j_{\text{in}}}{\pi R} \sum_{n=1}^{\infty} (1-(-1)^n)\, n\, (n+3)(n+4) \left(\frac{r}{R}\right)^{n-1} \left(1 - \frac{r^2}{R^2}\right)^2 \sin\left(\frac{\pi n}{3}\right) \cos(n\theta), \quad \text{(B.11a)}
$$

$$
j_\theta(r, \theta) = -\frac{4 j_{\text{in}}}{\pi} \sum_{m=1}^{\infty} (m+1) \left(\frac{r}{R}\right)^{2m-1} \left(1 - \frac{r^2}{R^2}\right) \sin\left(\frac{2\pi m}{3}\right) \cos(2m\theta)
$$

$$
+ \frac{\xi j_{\text{in}}}{\pi R} \sum_{n=1}^{\infty} (1-(-1)^n)(n+3)(n+4) \left(\frac{r}{R}\right)^{n-1}
$$

$$
\times \left(1 - \frac{r^2}{R^2}\right) \left(n - (n+4)\frac{r^2}{R^2}\right) \sin\left(\frac{\pi n}{3}\right) \sin(n\theta). \quad \text{(B.11b)}
$$

### B.3  Stream function in the hexagonal device

The full stream function in polar coordinates is given by

$$
\psi(r, \theta) = -\frac{2R j_{\text{in}}}{\pi \rho_0} \sum_{n=1}^{\infty} \frac{1}{n} \left(\frac{r}{R}\right)^{2n} \left(1 + n\left(1 - \frac{r^2}{R^2}\right)\right) \sin\left(\frac{2\pi n}{3}\right) \cos(2n\theta)
$$

$$
- \frac{\xi j_{\text{in}}}{\pi \rho_0} \sum_{n=1}^{\infty} (1-(-1)^n)(n+3)(n+4) \left(\frac{r}{R}\right)^{n+4} \sin\left(\frac{\pi n}{3}\right) \sin(n\theta)
$$

$$
- \frac{\xi j_{\text{in}}}{\pi \rho_0} \sum_{n=1}^{\infty} (1-(-1)^n)(n+3)(n+4) \left(\frac{r}{R}\right)^{n} \left(1 - 2\frac{r^2}{R^2}\right) \sin\left(\frac{\pi n}{3}\right) \sin(n\theta),
$$

$$
\text{(B.12)}
$$

which, in Cartesian coordinates is given (to lowest order in $x$ and $y$) by

$$
\psi(x, y) = -\frac{\sqrt{3} R j_{\text{in}}}{\pi \rho_0} \left(\frac{x^2 - y^2}{R^2}\right) \left(2 - \frac{x^2 + y^2}{R^2}\right) + O(r^6)
$$

$$
- \frac{20\sqrt{3}\, \xi j_{\text{in}}}{\pi \rho_0 R^5} (x^2 + y^2)^2\, y + O(r^9) \frac{20\sqrt{3}\, \xi j_{\text{in}}}{\pi \rho_0 R^3} (2x^2 + 2y^2 - R^2)\, y + O(r^5).
$$

$$
\text{(4.65)}
$$

## C  Details for channel flow

### C.1  Equations of motion

We start with the linearized continuity equations,

$$
0 = \partial_t \rho + \partial_i j_i, \tag{C.1a}
$$

$$
0 = \rho_0 \partial_t v_i + \partial_j \tau_{ji}, \tag{C.1b}
$$

which can be written explicitly as

$$
\partial_t \rho = -\rho_0 \partial_i v_i + D \partial^2 \rho + 2\alpha \partial_x \partial_y v_x + \alpha \left(\partial_x^2 - \partial_y^2\right) v_y, \tag{C.2a}
$$

$$
\rho_0 \partial_t v_x = -c^2 \partial_x \rho + \eta \partial^2 v_x + \zeta \partial_x \partial_j v_j - \eta_\circ \partial_y \partial_k \epsilon_{kl} v_l + \beta \lambda_{ijk} \partial_j \partial_k \rho, \tag{C.2b}
$$

$$
\rho_0 \partial_t v_y = -c^2 \partial_y \rho + \eta \partial^2 v_y + \zeta \partial_y \partial_j v_j + \eta_\circ \partial_x \partial_k \epsilon_{kl} v_l + \beta \lambda_{ijk} \partial_j \partial_k \rho. \tag{C.2c}
$$

## C.2 Rotated coordinates

In considering channel flow in Sec. 4.2, we must consider arbitrary orientations of the Fermi surface relative the channel walls. For concreteness, let us suppose one of the edges of the triangular Fermi surface aligns with the $x$ axis in (3.28).

The coordinate rotation is given by a matrix $R(\varphi)$ with $\tilde{x}_\mu = R_{\mu,j}(\varphi) x_j$ for the coordinates (and generic vectors) and $\partial_j = R^\dagger(\varphi)_{j,\mu} \partial_\mu = \partial_\mu R_{\mu,j}(\varphi)$, where we use Greek indices for the rotated coordinate frame. Explicitly, we have

$$\begin{pmatrix} \tilde{x} \\ \tilde{y} \end{pmatrix} = \begin{pmatrix} \cos\varphi & \sin\varphi \\ -\sin\varphi & \cos\varphi \end{pmatrix} \begin{pmatrix} x \\ y \end{pmatrix}, \quad \text{and} \quad \begin{pmatrix} \partial_{\tilde{x}} \\ \partial_{\tilde{y}} \end{pmatrix} = \begin{pmatrix} \cos\varphi & \sin\varphi \\ -\sin\varphi & \cos\varphi \end{pmatrix} \begin{pmatrix} \partial_x \\ \partial_y \end{pmatrix}. \tag{C.3}$$

Since the coordinate change leaves all but the $\mathrm{D}_6$ terms unchanged, we need only consider its action upon these two terms:

$$\begin{aligned} \lambda_{ijk}\partial_i\partial_j v_k &= \partial_i \sigma^x_{ij} \partial_j v_x + \partial_i \sigma^z_{ij} \partial_j v_y \\ &= \tilde{\nabla} \cdot R \cdot \sigma^x \cdot R^\dagger \cdot \tilde{\nabla}\left(\hat{e}_{\tilde{x}} \cdot R^\dagger \cdot \tilde{v}\right) + \tilde{\nabla} \cdot R \cdot \sigma^z \cdot R^\dagger \cdot \tilde{\nabla}\left(\hat{e}_{\tilde{y}} \cdot R^\dagger \cdot \tilde{v}\right), \end{aligned} \tag{C.4}$$

where we have

$$\tilde{\nabla} \cdot R \cdot \sigma^x \cdot R^\dagger \cdot \tilde{\nabla} = \sin(2\varphi)\left(\partial^2_{\tilde{x}} - \partial^2_{\tilde{y}}\right) + 2\cos(2\varphi)\partial_{\tilde{x}}\partial_{\tilde{y}}, \tag{C.5a}$$

$$\tilde{\nabla} \cdot R \cdot \sigma^z \cdot R^\dagger \cdot \tilde{\nabla} = \cos(2\varphi)\left(\partial^2_{\tilde{x}} - \partial^2_{\tilde{y}}\right) - 2\sin(2\varphi)\partial_{\tilde{x}}\partial_{\tilde{y}}, \tag{C.5b}$$

and therefore

$$\begin{aligned} \lambda_{ijk}\partial_i\partial_j v_k &= \cos(\varphi)\left[\sin(2\varphi)\left(\partial^2_{\tilde{x}} - \partial^2_{\tilde{y}}\right) + 2\cos(2\varphi)\partial_{\tilde{x}}\partial_{\tilde{y}}\right]v_{\tilde{x}} \\ &\quad - \sin(\varphi)\left[\sin(2\varphi)\left(\partial^2_{\tilde{x}} - \partial^2_{\tilde{y}}\right) + 2\cos(2\varphi)\partial_{\tilde{x}}\partial_{\tilde{y}}\right]v_{\tilde{y}} \\ &\quad + \cos(\varphi)\left[\cos(2\varphi)\left(\partial^2_{\tilde{x}} - \partial^2_{\tilde{y}}\right) - 2\sin(2\varphi)\partial_{\tilde{x}}\partial_{\tilde{y}}\right]v_{\tilde{y}} \\ &\quad + \sin(\varphi)\left[\cos(2\varphi)\left(\partial^2_{\tilde{x}} - \partial^2_{\tilde{y}}\right) - 2\sin(2\varphi)\partial_{\tilde{x}}\partial_{\tilde{y}}\right]v_{\tilde{x}}, \end{aligned} \tag{C.6}$$

and we can combine like terms to find

$$\begin{aligned} \lambda_{ijk}\partial_i\partial_j v_k &= \left\{\sin(3\varphi)\left(\partial^2_{\tilde{x}} - \partial^2_{\tilde{y}}\right) + 2\cos(3\varphi)\partial_{\tilde{x}}\partial_{\tilde{y}}\right\}v_{\tilde{x}} \\ &\quad + \left\{\cos(3\varphi)\left(\partial^2_{\tilde{x}} - \partial^2_{\tilde{y}}\right) - 2\sin(3\varphi)\partial_{\tilde{x}}\partial_{\tilde{y}}\right\}v_{\tilde{y}}, \end{aligned} \tag{C.7}$$

and likewise,

$$\begin{aligned} \lambda_{ijk}\partial_i\partial_j \rho &= \left\{\sin(3\varphi)\left(\partial^2_{\tilde{x}} - \partial^2_{\tilde{y}}\right) + 2\cos(3\varphi)\partial_{\tilde{x}}\partial_{\tilde{y}}\right\}\delta_{k,\tilde{x}}\rho \\ &\quad + \left\{\cos(3\varphi)\left(\partial^2_{\tilde{x}} - \partial^2_{\tilde{y}}\right) - 2\sin(3\varphi)\partial_{\tilde{x}}\partial_{\tilde{y}}\right\}\delta_{k,\tilde{y}}\rho, \end{aligned} \tag{C.8}$$

which we note is symmetric under $2\pi/3$ rotations (which preserve a triangular Fermi surface).

## C.3 Continuity equations

We now introduce an electric field $E$ oriented along the channel (in the $\tilde{x}$ direction) by replacing spatial derivatives of the density $\rho$ everywhere they appear according to

$$\partial_i \rho \rightarrow \partial_i \rho - \chi E_i, \tag{C.9}$$

where $E_{x,y}$ are the electric field components, and $\chi = \rho_0/c^2$ is the charge susceptibility.

Combining this with the rotated terms proportional to $\alpha$ and $\beta$, we now take $\partial_t \rho = \partial_t v_i = 0$ (the steady-state limit), giving new continuity equations

$$\rho_0 \tilde{\nabla} \cdot \tilde{v} = D \tilde{\nabla}^2 \rho + \alpha \left( \partial_{\tilde{x}}^2 - \partial_{\tilde{y}}^2 \right) \left( \sin(3\varphi) v_{\tilde{x}} + \cos(3\varphi) v_{\tilde{y}} \right) + 2\rho_0 \xi \, \partial_{\tilde{x}} \partial_{\tilde{y}} \left( \cos(3\varphi) v_{\tilde{x}} - \sin(3\varphi) v_{\tilde{y}} \right), \quad \text{(C.10a)}$$

$$c^2 \partial_{\tilde{x}} \rho = c^2 \chi E + \eta \tilde{\nabla}^2 v_{\tilde{x}} + \zeta \, \partial_{\tilde{x}} \left( \tilde{\nabla} \cdot \tilde{v} \right) - \eta_\circ \, \partial_{\tilde{y}} \left( \tilde{\nabla} \times \tilde{v} \right) + c^2 \xi \left[ \sin(3\varphi) \left( \partial_{\tilde{x}}^2 - \partial_{\tilde{y}}^2 \right) + 2 \cos(3\varphi) \partial_{\tilde{x}} \partial_{\tilde{y}} \right] \rho, \quad \text{(C.10b)}$$

$$c^2 \partial_{\tilde{y}} \rho = \eta \tilde{\nabla}^2 v_{\tilde{y}} + \zeta \, \partial_{\tilde{y}} \left( \tilde{\nabla} \cdot \tilde{v} \right) + \eta_\circ \, \partial_{\tilde{x}} \left( \tilde{\nabla} \times \tilde{v} \right) + c^2 \xi \left[ \cos(3\varphi) \left( \partial_{\tilde{x}}^2 - \partial_{\tilde{y}}^2 \right) - 2 \sin(3\varphi) \partial_{\tilde{x}} \partial_{\tilde{y}} \right] \rho, \quad \text{(C.10c)}$$

and we further restrict to solutions that do not vary with $\tilde{x}$: Setting all $\partial_{\tilde{x}}$ terms to zero gives

$$\rho_0 \partial_{\tilde{y}} v_{\tilde{y}} = D \, \partial_{\tilde{y}}^2 \rho - \alpha \sin(3\varphi) \partial_{\tilde{y}}^2 v_{\tilde{x}} - \alpha \cos(3\varphi) \partial_{\tilde{y}}^2 v_{\tilde{y}}, \quad \text{(4.68a)}$$

$$-c^2 \chi E = \left( \eta + \eta_\circ \right) \partial_{\tilde{y}}^2 v_{\tilde{x}} - \beta \sin(3\varphi) \partial_{\tilde{y}}^2 \rho, \quad \text{(4.68b)}$$

$$c^2 \partial_{\tilde{y}} \rho = (\eta + \zeta) \partial_{\tilde{y}}^2 v_{\tilde{y}} - \beta \cos(3\varphi) \partial_{\tilde{y}}^2 \rho, \quad \text{(4.68c)}$$

with the components of the current given by

$$j_{\tilde{x}} = \rho_0 v_{\tilde{x}} + D \chi E - \alpha \cos(3\varphi) \partial_{\tilde{y}} v_{\tilde{x}} + \alpha \sin(3\varphi) \partial_{\tilde{y}} v_{\tilde{y}}, \quad \text{(4.69a)}$$

$$j_{\tilde{y}} = \rho_0 v_{\tilde{y}} - D \partial_{\tilde{y}} \rho + \alpha \sin(3\varphi) \partial_{\tilde{y}} v_{\tilde{x}} + \alpha \cos(3\varphi) \partial_{\tilde{y}} v_{\tilde{y}}. \quad \text{(4.69b)}$$

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
