# Peer review of "Hydrodynamics with triangular point group"

_SciPost Physics, doi:SciPost Phys. 14, 137 (2023)_

## Round 2 · Referee Report · Anonymous (Referee 1) · 2022-10-16

Report

See appended pdf.

---

## Round 2 · Referee Report · Anonymous (Referee 2) · 2022-11-28

Report

The authors present a rather detailed discussion of a conjectured hydrodynamic theory of a D_6 electronic fluid. Two experiments in ultra-pure materials with D_6-invariant Fermi surfaces are suggested in order to detect new dissipative kinetic coefficients that are proposed in this work.

The paper is well-written and well-organized. I do however have a few questions mostly about what's not included in the manuscript.

The authors focus on symmetry properties (which is completely understandable in the context of hydrodynamics), but don't mention anything about microscopic interactions, both respecting and breaking the assumed symmetries. As a result, it is unclear how could one establish a parameter range (e.g., in temperature, magnetic field, etc.) where the proposed hydrodynamic behavior could be expected to be observable, as well as essential features such as the temerature and magnetic field dependence of transport coefficients. Similarly, the authors do not discuss details of the band structure that might affect the hydrodynamic behavior (similar to the striking difference between hydrodynamics in neutral and doped graphene), which is especially important in regards to the assumed Galilean invariance or lack thereof. Finally, there is not much discussion regarding specific materials where the proposed behavior could be expected apart for a very brief section on trilayer graphene.

The proposed behavior is clearly dependent on the choice of the D_6 symmetry. In the absence of specific experiments (recent, ongoing, or planned), the choice of the symmetry group is perhaps motivated by the result itself, but it would be nice if the authors commented on that as well - after all, there is quite a number of point groups that are not usually discussed in the literature.

---

## Round 3 · Referee Report · Anonymous (Referee 3) · 2023-3-6

Report
I believe that the authors have adequately responded to all comments and suggestions by the Referees. I recommend the paper to be published.

---

## Round 3 · Author Response

https://www.dropbox.com/s/i7hb652vwrehhoj/d6_fluid_refereeresponse.pdf?dl=0

---

## Editorial Decision

published